# Interleukin-9 production by type 2 innate lymphoid cells induces Paneth cell metaplasia and small intestinal remodeling

Chengyin Yuan[1], Aditya Rayasam[1], Alison Moe[1], Michael Hayward[2], Clive Wells[3], Aniko Szabo [4], Andrew Mackenzie [5], Nita Salzman [2] & William R. Drobyski [1,2,3] ✉

Paneth cell metaplasia (PCM) typically arises in pre-existing gastrointestinal (GI) diseases; however, the mechanistic pathway that induces metaplasia and whether PCM is initiated exclusively by disorders intrinsic to the GI tract is not well known. Here, we describe the development of PCM in a murine model of chronic myelogenous leukemia (CML) that is driven by an inducible bcr-abl oncogene. Mechanistically, CML induces a proinflammatory state within the GI tract that results in the production of epithelial-derived IL-33. The binding of IL-33 to the decoy receptor ST2 leads to IL-9 production by type 2 innate lymphoid cells (ILC2) which is directly responsible for the induction of PCM in the colon and tissue remodeling in the small intestines, characterized by goblet and tuft cell hyperplasia along with expansion of mucosal mast cells. Thus, we demonstrate that an extra-intestinal disease can trigger an ILC2/IL-9 immune circuit, which induces PCM and regulates epithelial cell fate decisions in the GI tract.

Paneth cells are specialized secretory epithelial cells that are generated from leucine-rich-repeat-containing G-protein-coupled receptor 5 (Lgr5) positive stem cells and are located at the bottom of crypts in the small intestines[1,2]. Signaling through the wnt pathway induces the subsequent maturation of these cells within intestinal crypts[3]. Paneth cells produce large quantities of antimicrobial peptides (AMPs), inflammatory mediators, and signaling molecules[4] and hence play a pivotal role in maintaining a homeostatic balance between commensal bacteria and the host immune system[5]. The production of AMPs, which include α-defensins, lysozyme and secretory phospholipase A2 (sPLA2), are particularly important for the maintenance of intestinal immune homeostasis and the shaping of microbial composition within the small intestines[6]. Of the AMPs produced by Paneth cells, α-defensins (also called crypt defensins or cryptdins in the mouse) are the most abundant of these secretory proteins[7]. They are synthesized as inactive propeptides and then activated after cleavage by metalloproteinase 7 where they regulate the microbial ecosystem[6].

Numerous studies have shown that the disruption of Paneth cell function plays a key role in many diseases such as inflammatory bowel disease and graft versus host disease[8–11]. The damage to gut epithelial and Paneth cells in these disease states results in the loss of AMP production as well as a reduction in microbial diversity. Interestingly, in some disease settings, Paneth cells can also appear outside of the natural environment of the small intestines, a phenomenon which has been termed Paneth cell metaplasia (PCM). Sites of metaplasia are typically in the colon, stomach, and esophagus and most often are associated with pathological conditions involving the GI tract, such as inflammatory bowel disease, Barrett's esophagus, and colorectal cancer[12–15]. A hallmark of PCM is that it typically arises in the setting of chronic inflammation and pathological damage in the GI tract; however, the mechanistic pathway by which PCM develops under these conditions remains unknown.

In the current study, we report the development of PCM in a murine model of chronic myelogenous leukemia (CML) which is driven

[1]Department of Medicine, Medical College of Wisconsin, Milwaukee, WI, USA. [2]Department of Pediatrics, Medical College of Wisconsin, Milwaukee, WI, USA. [3]Department of Microbiology and Immunology, Medical College of Wisconsin, Milwaukee, WI, USA. [4]Division of Biostatistics, Institute of Health and Equity, Medical College of Wisconsin, Milwaukee, WI, USA. [5]MRC Laboratory of Molecule Biology, Cambridge, UK. ✉e-mail: wdrobysk@mcw.edu

by the hematopoietic stem cell-restricted expression of the bcr/abl oncogene. Mechanistically, PCM is attributable to intestinal epithelial cell-induced IL-33 production that arises in the setting of CML-induced gastrointestinal (GI) tract inflammation. This leads to the activation of type 2 innate lymphoid cells (ILC2s) resulting in interleukin-9 (IL-9)-driven PCM, mucosal mast cell hyperplasia, and extensive remodeling of the small intestine characterized by marked lengthening along with goblet and tuft cell hyperplasia. Thus, intestinal inflammation can trigger an ILC2/IL-9 circuit that induces metaplasia in Paneth cells and regulates the development of specialized epithelial cell populations resident in the GI tract.

## Results

### Paneth cell metaplasia emerges during bcr/abl oncogene-dependent CML

We employed a murine bcr-abl oncogene-dependent CML model that recapitulates the development and clinical manifestations of human disease[16]. Specifically, cross-breeding of SCLtTA and bcr-abl mice results in double transgenic animals in which expression of the bcr-abl oncogene and the subsequent development of CML is regulated by tetracycline (Tet) administration (Supplementary Fig. 1a). To standardize disease kinetics within a reproducibly defined temporal window, we developed a murine CML transplantation model[17] in which lethally irradiated wild type FVB mice are transplanted with bone marrow (BM) cells from SCLtTA/bcr-abl animals (henceforth referred to as CML mice) and then maintained on or off Tet. Recipients maintained off Tet exhibited weight loss (Supplementary Fig. 1b), along with a significant temporal increase in white blood cells (Supplementary Fig. 1c) and granulocytes (Supplementary Fig. 1d) in the peripheral blood. These animals also had a nearly two-fold augmentation in spleen weights (Supplementary Fig. 1e) and an increase in the frequency (Supplementary Fig. 1f) and absolute number of CD11b$^+$ Gr-1$^+$ myeloid cells (Supplementary Fig. 1g) consistent with what is observed in CML in humans[18,19]. There was also a loss of normal follicular architecture in the spleen and mesenteric lymph nodes due to the expansion of myeloid cells in these sites (Supplementary Fig. 1h, i). In addition, histological analysis of non-lymphoid tissues revealed accumulation of neutrophils in the liver and lung of leukemic animals (Supplementary Fig. 1j, k).

Unexpectedly, during the histological examination of the colon, we observed that CML mice had a prominent number of cells with cytoplasmic eosinophilic granules that were morphologically consistent with Paneth cells (Fig. 1a). In addition, whereas Paneth cells in the ileum of non-leukemic animals were located appropriately in the basal region of crypts, Paneth cells in the ileum of CML mice were scattered throughout the villi (see arrows Fig. 1b). Staining of colonic tissue with phloxine tartrazine revealed characteristic Paneth cell granule affinity for phloxine (Fig. 1b). In addition, electron microscopy also confirmed that these were Paneth cells as evidenced by the presence of spherical electron dense granules (Fig. 1c) and tight junctions which identified these cells as epithelial and not haematopoietically-derived (Fig. 1d). Paneth cells in the colon and ileum also stained positively for the secretory anti-microbial protein lysozyme (Fig. 1e) and for MPTX 1/2 in the ileum (Supplementary Fig. 2). We observed that there was a significant increase in the absolute number of Paneth cells in both the colon and ileum when compared to animals maintained on Tet alone (Fig. 1f, g), indicative of Paneth cell metaplasia (PCM) in the colon and hyperplasia in the ileum. Analysis of antimicrobial peptide gene expression revealed a progressive and statistically significant increase in mRNA levels of cryptdin 1 and sPLA2 in CML mice, demonstrating Paneth cell-specific gene expression (Fig. 1h, i). To prove a direct and causal relationship between the development of CML and PCM, we exploited the fact that the administration of Tet in the drinking water to mice with clinically evident CML reverses the leukemia phenotype and normalizes the WBC count in the peripheral blood (see Experimental

Approach Fig. 1j). Transplant recipients in both cohorts were maintained off Tet for the first 35 days at which point the majority develop leukocytosis (Fig. 1k). Tet administration on day 35 to mice with leukocytosis and previously documented PCM (Fig. 1f) normalized the WBC within three weeks, whereas animals that remained off Tet had progressive leukocytosis (Fig. 1k). Mice that had normalized their WBC counts also had resolution of PCM, whereas animals with CML and off Tet had persistent metaplasia (Fig. 1l, m). Thus, there was a direct and causal link between the temporal development of CML and PCM within the colon in murine transplant recipients.

### IL-9 drives Paneth cell metaplasia in CML mice

To delineate a mechanistic pathway by which PCM develops in these mice, we hypothesized a role for IL-9 based on a report which had demonstrated that over-expression of this cytokine in a non-disease transgenic mouse model induced PCM in the colon[20]. To that end, we observed a significant increase in IL-9 gene expression in the mesenteric lymph nodes, ileum, and colons of animals with CML, but not in the bone marrow or spleen (Fig. 2a). Conversely, expression of IL-9 in non-leukemic (Tet ON) mice was essentially undetectable in all these tissue sites. IL-9 receptor mRNA levels were also augmented in the ileum and colon, but significantly reduced in the spleen of CML animals likely due to the expansion of non-IL-9R-expressing granulocyte populations (Fig. 2b). To assess the functional role of IL-9 in the induction of PCM, we treated mice with an anti-IL-9 antibody and observed that these animals had increased weight loss (Supplementary Fig. 3a) and a significant time-dependent increase in white blood cell and granulocyte counts in the peripheral blood when compared to mice that received an isotype control antibody (Supplementary Fig. 3b, c). While there was no difference in spleen weight between groups (Supplementary Fig. 3d), the absolute number of myeloid CD11b$^+$ Gr-1$^+$ cells were also significantly augmented in mice that were treated with anti-IL-9 antibody (Supplementary Fig. 3e), demonstrating that IL-9 blockade increased the absolute number of granulocytes in CML mice. With respect to the development of PCM, the administration of an anti-IL-9 antibody to CML mice significantly reduced the absolute number of Paneth cells in the colon, but not the ileum, indicating that PCM was regulated by IL-9 (Fig. 2c, d). Immunofluorescence images also demonstrated a reduction in Paneth cells in the colon, but not the ileum in these animals (Fig. 2e). Anti-IL-9 antibody-treated mice had a corresponding decrease in mRNA levels of cryptdin 1 and sPLA2 in the colon and ileum when compared to CML mice that received an isotype control antibody, as well as animals maintained on Tet (Fig. 2f, g). Thus, blockade of IL-9 signaling effectively abrogated the development of PCM in the colon and reduced gene expression of Paneth cell AMPs in both the ileum and colon of leukemic animals.

### Metabolic profiling identifies histamine metabolites in the GI tract

The unexpected emergence of PCM in the colon and Paneth cell hyperplasia in the ileum of CML mice led us to examine whether other cellular and metabolic alterations occurred in the GI tract of these animals. To address this question, we performed unbiased metabolic profiling and examined 973 biochemical metabolites in ileal tissue from leukemic (CML) and non-leukemic FVB recipients of syngeneic marrow grafts (FVB SYN). A total of 676 metabolites were differentially expressed ($p < 0.05$ cutoff) between these two cohorts with 292 metabolites overexpressed and 384 under-expressed in CML mice (Fig. 3a). Principal component analysis confirmed that CML mice differed substantially from non-leukemic mice with over 50% of the observed variance demonstrable along PCA1 (Fig. 3b). Hierarchical clustering analysis demonstrated that biochemical differences were most prominently observed in amino acid and phospholipid metabolism (Fig. 3c). A more discriminatory and statistically rigorous metabolic analysis revealed that there were 59 overexpressed and 78 under-

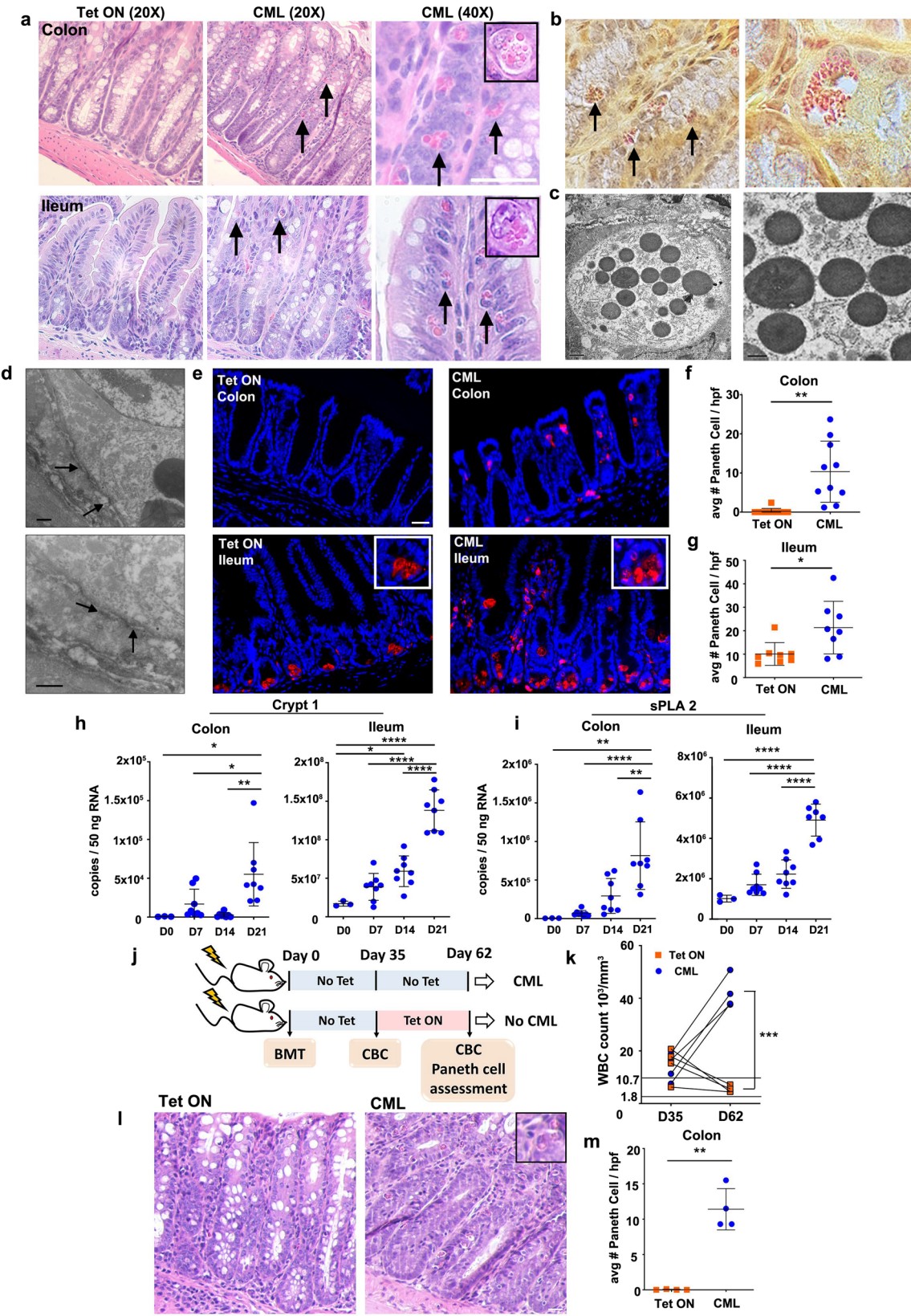

expressed metabolites between these two cohorts based on defined cutoff criteria (|log$_2$(fold change)| >1.0 and $p_{adjusted}$ < 0.0001 (Fig. 3d), full list available in Supplementary Data 1). Examination of the top twenty most over-expressed metabolites in CML mice revealed that three of the top 10 were involved in the metabolism of the aromatic amino acid histidine. Specifically, histamine which is directly

downstream from histidine by way of the enzymatic action of histidine decarboxylase was the most highly expressed metabolite (144-fold increase) (Fig. 3e). Furthermore, the downstream histamine metabolites, 1-methyl-4-imidazoleacetate and 1-methylhistamine (Fig. 3f), were also significantly higher in CML versus non-leukemic syngeneic controls (Fig. 3g), indicating that there was activation of the histamine

**Fig. 1 | Paneth cell metaplasia (PCM) develops during CML. a**–**i** Lethally irradiated FVB mice were transplanted with BM from SCLtTa/bcr-abl animals, and maintained on or off tetracycline (Tet). **a** Representative hematoxylin and eosin-stained sections of colon and ileum from animals on or off Tet (CML) 39 days post-transplantation. Original magnification is ×20 and ×40. Arrows denote Paneth cells. Inset (magnification ×60) depicts Paneth cells. **b** Phloxine/tartrazine staining of colon from CML animals. Original magnification is ×40 (left panel) and ×60 (right panel). **c**, **d** Electron microscopy photomicrograph of Paneth cells in the colon of CML mice showing secretory granules (**c**) and tight junctions (see arrows) between Paneth cell (upper panel, granules visible) and adjacent cell (**d**). Magnification is ×3000 and ×15,000 for (**c**), and ×20,000 and ×35,000 for (**d**). Scale bar is 2 μm and 500 nm for (**c**) and 500 nm for (**d**). **e** Immunofluorescence staining of lysozyme in Paneth cell granules in the colon and ileum. Magnification is ×20. Insets show granules in the ileum. **f**, **g** Average number of Paneth cells per 10 high power fields in the colon and ileum 34–39 days post-transplantation. Results are from two

experiments (n = 8–11 mice/group). **h, i** Crypt1 and sPLA2 gene expression in the colon and ileum of CML mice. Data are from two experiments (n = 3–8 mice/group). **j**–**m** FVB mice were transplanted with BM from SCLtTa/bcr-abl animals and maintained off Tet for 35 days. On day 35, one cohort was placed back on Tet, while the other remained on normal water. **j** Experimental scheme. **k** White blood cell counts. Data are from one experiment (n = 4 mice). **l** Photomicrograph depicting PCM in animals maintained off Tet and resolution of PCM in mice placed back on Tet. Inset shows Paneth cell in CML animal. **m** Average number of Paneth cells 62 days post-transplantation. Results are from one of two experiments that yielded similar results (n = 4 mice/group). Scale bar = 30 μm for all other panels. Data are shown as mean ± SD. Statistics were performed using Welch's t test (pairwise comparisons) and one-way ANOVA plus Fishers LSD test (greater than three comparisons). *p < 0.05, **p < 0.01, ***p < 0.001, ****p < 0.0001. Source data are provided as a Source Data file.

pathway in the GI tract that occurred concurrently with the development of PCM.

## CML induces IL-9-regulated hyperplasia of specialized intestinal epithelial cells

Histamine is found solely in basophils and mast cells in the periphery; however, only mast cells are tissue resident[21]. The histamine pathway metabolite differences that we identified in CML mice therefore prompted us to examine whether mast cells were increased in the GI tract of these animals, and if their presence was similarly regulated by IL-9. This premise was supported by the observation that ileal tissue from CML animals had a ten-fold increase in serotonin which is also produced by mast cells (Figs. 3e, 4a)[22]. Gene expression of the IgE high-affinity receptor (Fcer1a) which is present on all mast cell populations was significantly increased in the ileum of CML mice and correspondingly reduced in anti-IL-9 antibody-treated animals (Fig. 4b). Mast cells are present in two different locations (i.e., connective tissue and the epithelial layer) which can be distinguished by specific mast cell proteases[23]. To that end, we found that expression of mMCP1,2,4, which is produced by both mucosal and connective tissue mast cells, was significantly increased in leukemic animals but reduced in anti-IL-9 antibody-treated mice (Fig. 4c), whereas mMCP-7 which is produced by connective tissue mast cells was reduced in CML mice (Fig. 4d). Isolation of intestinal epithelial (IECs) and lamina propria cells from the colon revealed that mMCP 1,2,4 was predominantly expressed in IECs, confirming that mast cells were confined to the mucosa (Fig. 4e). Immunohistochemistry staining with mMCP 1, which is mucosally restricted, also demonstrated mast cell localization in the epithelial layer of the ileum in CML mice (Fig. 4f).

In conjunction with mast cell alterations, we observed that there was CML-induced intestinal remodeling as evidenced by a marked increase in small intestinal, but not colonic, length that was partially attenuated by IL-9 signaling blockade (Fig. 4g). BrdU labeling demonstrated that small intestinal lengthening was associated with increased proliferation in the ileal crypts of CML mice (Fig. 4h, i). This phenomenon has been observed in helminth infections where tuft cells respond to succinate and drive intestinal remodeling in infected hosts[23]. In that regard, metabolic profiling studies revealed that there were increased succinate levels (Fig. 4j), as well as augmented gene expression of the succinate receptor (sucnr1) in the ileum of CML mice (Fig. 4k). Furthermore, expression of the Tuft cell marker DCLK-1 (double cortin-like kinase-1) and IL-25, which is secreted by Tuft cells and regulates type 2 immune responses[24,25], increased in a time-dependent manner (Fig. 4l) that coincided with the temporal kinetics of leukemia progression (Supplementary Fig. 1). Gene expression of DCLK-1 and IL-25 in the ileum of CML mice (Fig. 4m) as well as the absolute number of DCLK-1+ tuft cells were also significantly reduced by IL-9 signaling blockade (Fig. 4n, o). Concurrent with tuft cell hyperplasia, we

observed that CML mice had an increased mucus layer that was present in the ileum (Fig. 4p). In addition, these animals also had goblet cell hyperplasia that was significantly reduced in anti-IL-9 antibody-treated animals (Fig. 4q, r). Expression of the goblet cell-specific gene, Gob-5, which is a calcium-activated chloride gene involved in mucin production[26], was augmented in leukemic animals, although expression was not affected by IL-9 signaling blockade (Fig. 4s). Collectively, these studies demonstrated that CML induced extensive intestinal remodeling characterized by the IL-9 regulated expansion of mucosal mast, tuft and goblet cells.

## IL-13 regulates tuft and goblet cell hyperplasia, but not PCM during CML

Binding of succinate to the succinate receptor on tuft cells leads to the secretion of IL-25 which activates ILC2s leading to release of IL-13[27]. IL-13 is then able to promote the renewal of intestinal stem cells and their differentiation into tuft and goblet cells resulting in a feed forward loop[28]. The observation that succinate was increased in the ileum of mice with CML therefore led us to examine whether IL-13, in addition to IL-9, co-regulated the expansion of specialized intestinal epithelial cell populations. We observed that IL-13 mRNA levels were significantly increased in the ileum, colon, mesenteric lymph nodes, and bone marrow of leukemic animals with differential expression in the ileum being the most pronounced (Fig. 5a). In fact, the gene expression profile of IL-13 was similar to that of IL-9 in CML versus non-leukemic mice with the exception of the bone marrow (Fig. 2a). Administration of an anti-IL-13 antibody that had previously been shown to effectively inhibit allergic disease in mice[29] had no effect on PCM or hyperplasia in either the colon or ileum, respectively (Fig. 5b). There was, however, a significant reduction in crypt 1 (Fig. 5c) but not sPLA2 (Fig. 5d) gene expression in the ileum and colon of animals that received anti-IL-13 antibody. Conversely, blockade of IL-13 signaling had no effect on gene expression of the mast cell markers Fcer1-a, mMCP 1,2,4 or mMCP 7 (Fig. 5e–g), indicating that IL-9, and not IL-13, was the primary regulator of mast cell gene expression. Administration of anti-IL-13 antibody, however, did result in a significant reduction in the expression of DCLK-1 and IL-25 in tuft cells (Fig. 5h) and Gob-5 in goblet cells (Fig. 5i). In contrast, examination of IL-5, another prototypical $T_H2$ cytokine, demonstrated no effect of IL-5 cytokine blockade on gene expression of IL-9 or IL-13 in the ileum (Supplementary Fig. 4a, b), expression of crypt1 or sPLA in the ileum and colon (Supplementary Fig. 4c, d), or expression of mMCP 1,2,4 in mast cells or DCLK-1 and IL-25 in tuft cells (Supplementary Fig. 4e, f). Thus, these studies demonstrated that IL-13 regulated Paneth cell-specific gene expression along with tuft cell and goblet cell hyperplasia, although this cytokine had no effect on the development of Paneth cell metaplasia or hyperplasia during CML.

To determine whether IL-25, whose expression was increased in the ileum of CML mice (Fig. 4l, m), regulated the production of

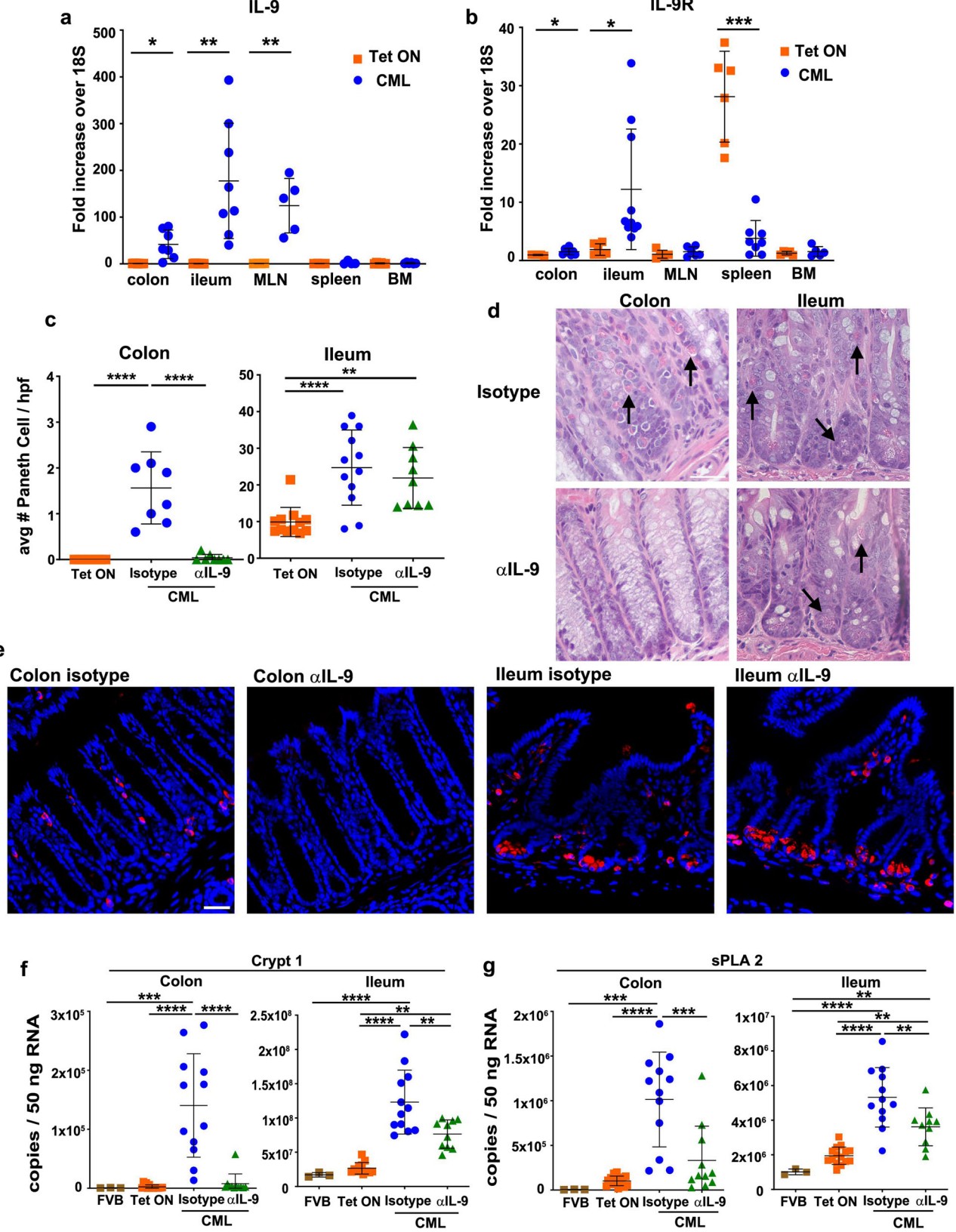

IL-13 as well as Paneth and specialized epithelial cell gene expression, we treated these mice with a blocking anti-IL-25 or isotype control antibody. These experiments revealed that inhibition of IL-25 signaling had no effect on gene expression of IL-13 or IL-9 in the ileum (Fig. 5j) nor was there any difference in the expression of crypt1 or sPLA in the ileum or colon of anti-IL-25 treated animals

except for crypt1 in the colon (Fig. 5k, l). In addition, antibody blockade of IL-25 did not alter gene expression of mMCP 1,2,4 in mast cells or DCLK-1 in tuft cells (Fig. 5m, n), indicating that IL-25 had no apparent role in the upstream regulation of IL-13, or the expression of genes associated with specialized epithelial cell populations in CML mice.

**Fig. 2 | Paneth cell metaplasia is IL-9 dependent. a–f** Lethally irradiated FVB mice were transplanted with BM cells from SCLtTa/bcr-abl animals. **a, b** mRNA expression of IL-9 (**a**) and IL-9 receptor (**b**) in the colon, ileum, mesenteric lymph nodes, spleen, and bone marrow of animals 21-22 days post-transplantation that were maintained on or off Tet (CML). Data are expressed as fold increase over 18S. Results are from two–four experiments (*n* = 4–10 mice/group). **c** The average number of Paneth cells per 10 high power fields as determined by H&E staining in the colon and ileum of mice that were maintained on or off Tet and then treated three times per week with an isotype control or anti-IL-9 antibody is shown. Analysis was performed 21–22 days post-transplantation. Data are from two-three experiments (*n* = 8–12 mice/group). **d** Representative hematoxylin and eosin-stained sections of the colon and ileum from animals that were treated with an isotype control or anti-IL-9 antibody as in (**c**). Original magnification is ×40.

**e** Immunofluorescence staining of lysozyme in Paneth cell granules present in the colon and ileum of CML animals treated with an isotype control or anti-IL-9 antibody. Magnification is ×20. **f, g** Crypt1 and sPLA2 gene expression presented as copies/50 ng RNA in the colon and ileum of animals that were maintained on or off Tet and then treated with an isotype control or anti-IL-9 antibody three times per week. Normal nontransplanted FVB animals served as an additional control. Analysis was performed 21–22 days post-transplantation. Results are from three experiments (*n* = 3–15 animals/group). Data are presented as mean ± SD. Statistics were performed using Welch's *t* test (pairwise comparisons), one-way ANOVA plus Fishers LSD test (greater than three comparisons), and one-way ANOVA plus Tukey's correction (greater than three comparisons). *$p < 0.05$, **$p < 0.01$, ***$p < 0.001$, ****$p < 0.0001$. Source data are provided as a Source Data file.

## Type 2 innate lymphoid cells (ILC2s) are the primary producers of IL-9

To define the cellular source of IL-9, we considered that previous studies had shown IL-9 is produced by a variety of cell types, including type 2 helper T cells[30,31], T$_H$17 cells[32,33], CD4[+] regulatory T cells[34], innate lymphoid cells[35], and mast cells[31]. We therefore conducted an iterative approach in which we examined each of these cell populations to determine whether IL-9 production was detectable in these immune subsets. With respect to mast cells, while we had previously shown that mMCP 1,2,4 was predominantly expressed in the epithelial layer within the colon (Fig. 4e), expression of IL-9 was confined to the lamina propria of CML mice (Fig. 6a). In addition, immunofluorescence staining of colonic tissue revealed that there was no colocalization of mMCP-1 and IL-9 (Fig. 6b), indicating that mast cells were not the source of this cytokine within the GI tract. To determine whether T cells produced IL-9, we first examined the mLN since T cells are a prominent cellular component of this tissue and prior studies had shown increased expression of IL-9 in this tissue site (Fig. 2a). mLN cells were flow sorted into three populations (i.e., CD4[−] TCRβ[−], CD4[+] TCRβ[+] and CD4[−] TCRβ[+] cells) and q-PCR was employed to demonstrate that IL-9 expression was significantly increased in CD4[−] TCRβ[−] cells when compared to CD4[+] TCRβ[+] and CD4[−] TCRβ[+] T cells (Fig. 6c). Furthermore, immunofluorescence staining demonstrated a lack of co-localization of CD3 and IL-9 in both the ileum and colon (Fig. 6d), indicating that T cells were not the source of IL-9. As additional confirmation, we treated CML mice with an anti-CD4 antibody that depletes these T cells from tissue sites (Supplementary Fig. 5a). We observed no difference in IL-9 expression in the ileum and an increase in IL-9 expression in the colons of anti-CD4 antibody-treated mice (Fig. 6e). There was a modest decrease in the absolute number of Paneth cells in the colon but no difference in the ileum (Supplementary Fig. 5b), nor was there a difference in the expression of the Paneth cell genes, crypt 1 and sPLA 2, in the colon or ileum with the exception of crypt 1 in the colon (Supplementary Fig. 5c, d). Furthermore, there was no difference in colonic or small intestinal length (Supplementary Fig. 5e), or expression of mMCP 1,2,4, DCLK-1, and IL-25 in the ileum (Supplementary Fig. 5f–h). In contrast, nearly all IL-9-expressing cells in the ileum and colon co-expressed GATA3 (Fig. 6f), and the majority also had colocalized expression of KLRG1 (Fig. 6g), which are markers for ILC2s[36]. Quantification of IL-9 expressing cells in the ileum and colon from mice with CML revealed that an average of 92% and 95% of all IL-9[+] cells in the ileum and colon, respectively, were CD3[−]GATA3[+], providing further confirmation that these cells were ILC2s (Fig. 6h). Flow cytometry also demonstrated an increased percentage of ILC2s in the mLN and the colon (Fig. 6i). Furthermore, we observed increased expression of ST2 (*Il1rl1*) and IL-17RB (Fig. 6j, k), as well as IL-4 and IL-5 which are cytokines produced by ILC2s[37] (Fig. 6l–n). Collectively, these data identified type 2 innate lymphoid cells (ILC2s) as the primary source of IL-9 within the ileum and colon of CML mice.

## CML induces the production of intestinal epithelial-derived IL-33

We then sought to define upstream events that induced the ILC2-mediated production of IL-9 in the GI tract of CML mice. Using a semi-quantitative histological scoring system, we did not observe any difference in pathological scores obtained from colon samples of CML mice when compared to non-leukemic control animals (Fig. 7a). However, despite the lack of overt pathology, we did observe a significant increase in the percentage of neutrophils, macrophages and dendritic cells in the colons of CML mice, with neutrophilic accumulation being most pronounced and consistent with the granulocytosis that is characteristic of CML (Fig. 7b). Furthermore, CML also induced an inflammatory environment in the GI tract characterized by increased expression of IL-6, IFN-γ, TNF-α, IL-22, and GM-CSF in the ileum and colon (Fig. 7c). In addition, there were augmented levels of inflammatory lipids; specifically, eicosanoids (5-HETE, 12-HETE), glycerophospholipids (glycerophosphoserine, glycerophosphoinositol)[38] and mead acid[39] in the ileum of leukemic animals (Supplementary Data 1, Figs. 3e and 7d). To determine whether this inflammatory milieu was associated with cellular damage, we examined cleaved caspase 3 expression as a marker of apoptotic cell death. These studies revealed that there was increased expression of cleaved caspase 3 in the ileum and colons of CML when assessed by immuno-fluorescence staining (Fig. 7e) as well as by western blot analysis (Fig. 7f, g). High power images revealed a lack of co-localization of E-cadherin which resided in the cytoplasm and cleaved caspase 3 that was present in the nucleus of intestinal epithelial cells from CML animals (Supplementary Fig. 6). Prior studies have shown that IL-33, IL-25, and/or thymic stromal lymphopoietin (TSLP) are major activators of ILC2s[40], and these cytokines function as alarmins that are produced in response to cellular injury and inflammation[41]. Given the cell death observed in the intestines of CML mice, we examined gene expression of these cytokines and observed that mRNA expression of IL-33 was significantly increased in the ileum and nearly significant in the colon (Fig. 7h). Conversely, IL-25 which was previously shown to be augmented in the ileum where there was tuft cell hyperplasia (Fig. 4l), was not increased in the colon (Supplementary Fig. 7a) and TSLP was decreased in both tissue sites (Supplementary Fig. 7b). IL-33 is localized in the nucleus prior to its secretion as a 266 amino acid protein[42] that is subsequently cleaved by proteases from neutrophils and mast cells into smaller mature forms that have increased potency[43,44]. Western blot analysis confirmed increased expression of IL-33 in the ileum and colon of CML animals which was attributable to the more potent shorter 15 kDA cleavage protein (Fig. 7i, j, Supplementary Fig. 7c, e, f, h). Notably, IL-33 localized to the intestinal epithelial cell layer, as demonstrated by both western blot (Fig. 7k, l, Supplementary Fig. 7k) and immunofluorescence staining (Supplementary Fig. 7i, j), indicating that CML induced the production of epithelial-derived IL-33.

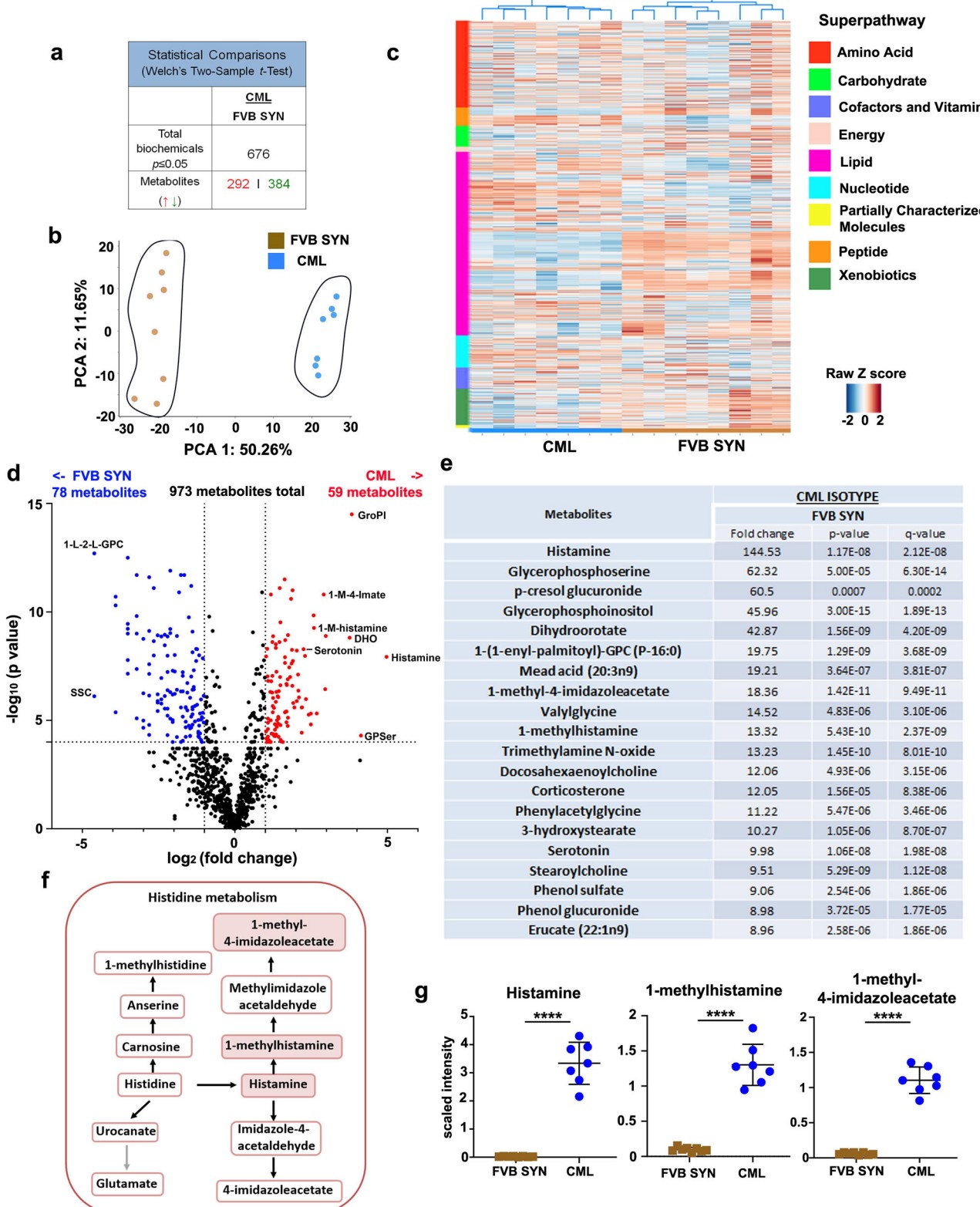

**g**

Histamine | 1-methylhistamine | 1-methyl-4-imidazoleacetate (plots, **** significance between FVB SYN and CML)

## IL-33 regulates the expression of ILC2-derived cytokines in the GI tract

To further define the upstream role of IL-33 and provide rationale for the predominance of more potent, smaller forms of this cytokine in CML mice, we examined proteases that have been shown to cleave IL-33 in vitro[43,44]. We observed the expression of neutrophil elastase (Fig. 8a, b, Supplementary Fig. 8a, c, d, f) and cathepsin G (Fig. 8a, c,

Supplementary Fig. 8b, c, e, f), as well as the mast cell protease, chymase (Fig. 8d, e, Supplementary Fig. 7d, e, g, h), were all significantly increased in the ileum and colon of CML animals, providing a potential explanation for the observed smaller IL-33 cleavage product[44]. Since IL-9 has a pivotal role in promoting PCM and intestinal remodeling, we then examined the effect of IL-9 signaling blockade on the production of IL-33 as well as proteases from these innate immune populations.

**Fig. 3 | Histamine and its downstream metabolites are over expressed in the GI tract of CML mice with PCM. a–g** Lethally irradiated FVB mice were transplanted with BM cells from normal FVB (syngeneic, FVB Syn) or SCLtTa/bcr-abl (CML) donors ($n = 7–8$ mice/group). Ileal tissue was harvested from animals 35 days post-transplantation and processed for metabolite analysis (see Material and Methods). **a** Total number of metabolites significantly differed ($p < 0.05$) between FVB SYN and CML mice. **b** Principal component analysis of experimental groups. PC1 (x-axis) accounted for 50.3% of variability while PC2 (y-axis) accounted for 11.7%. **c** Heat map depicting z-scored expression of metabolites which segregate into specified pathways in each of the experimental groups. Each pathway is color-coded and shown in the far-left column, while experimental groups are depicted in the bottom horizontal color-coded lines. **d** Volcano plot demonstrating differentially expressed metabolites between syngeneic recipients and CML mice. Cutoff parameters were |$\log_2$ fold change| > 1.0 and $p_{adjusted} < 0.0001$. **e** List of the top 20 metabolites that were over-expressed in CML mice versus syngeneic recipients. Fold-change is depicted along with associated p-values and q-values. Q values denote the false discovery rate and were computed using the method of Storey J and Tibshirani R. (see ref. 57). GroPI glycerophosphoinositol, GPSer glycerophosphoserine, DHO Dihydroorotate, SSC Cysteine s-sulfate, 1-L-2-L-GPC 1-linoleoyl-22-linoleoyl-GPC. **f** Graphical representation of the histidine metabolic pathway. **g** Scaled intensity of histamine, 1-methylhistamine, and 1-methyl-4-imidazoleacetate in each experimental group. Data are shown as mean ± SD. Statistics were performed using Welch's t test. ****$p < 0.0001$. Source data are provided as a Source Data file.

Administration of anti-IL-9 antibody had no effect on the 35 kDA product but did result in a significant decrease in the 15 kDA cleavage protein in both the ileum and colon (Fig. 8f–h, Supplementary Fig. 8n, o, p, q). Inhibition of IL-9 signaling reduced expression of neutrophil elastase only in the colon (Fig. 8f, i, Supplementary Fig. 8g, i, j, l) and had no effect on cathepsin G expression in the ileum or colon (Fig. 8f, j, Supplementary Figs. 8h, i, k, l) In contrast, expression of chymase was significantly lower in both the ileum and colon of mice treated with anti-IL-9 antibody (Figs. 8f and 8 Supplementary Figs. 8m, o, q), suggesting that mast cell-derived chymase was the primary mediator of proteolytic cleavage of IL-33. Finally, to assess the functional role of IL-33 and determine if this cytokine regulated the expression of IL-9 and other ILC2-derived cytokines, CML mice were treated with either an anti-ST2 or isotype control antibody. Inhibition of IL-33 signaling resulted in a significant decrease in gene expression of IL-9 (Fig. 8l), along with IL-4, IL-5 and IL-13 (Fig. 8m), but not IL-25 or mMCP (Fig. 8n) in the ileum of leukemic animals, demonstrating that IL-33 regulated the expression of ILC2-derived cytokines.

## Discussion

Paneth cell metaplasia (PCM) invariably occurs in diseases of the GI tract, such as inflammatory bowel disease and colorectal cancer[12–15]. In addition, parasitic infections within the intestines have been shown to induce metaplasia[45,46]; however, extra intestinal disorders have not been associated with this phenomenon. More importantly, the mechanistic pathway by which PCM develops in the GI tract has not been clearly delineated in any disease state. In the current study, we made the unexpected observation that CML, which is a hematological malignancy that is largely confined to the bone marrow, peripheral blood, and spleen, induced the development of PCM within the colon. To validate a causative relationship between CML and PCM, we employed a tetracycline-inducible mouse model that faithfully recapitulates the clinical manifestations associated with CML where mice develop leukocytosis, granulocytic hyperplasia, splenomegaly, and bcr-abl oncogene expression, all of which are hallmarks of this disease in humans[19]. We then exploited the fact that administration or withdrawal of tetracycline allowed us to temporally regulate bcr-abl oncogene expression and thereby leukemia onset and progression in a very precise manner. Consequently, we were able to demonstrate that the onset of PCM was coincident with the development of CML, and conversely, cessation of bcr-abl oncogene expression in hematopoietic stem cells by reinstitution of tetracycline resulted in the concurrent resolution of leukocytosis and metaplasia. Thus, these data provided a definitive and causal link between the temporal development of CML and PCM within the colon.

In addition to PCM, we observed extensive intestinal remodeling characterized by small intestinal lengthening along with tuft cell and goblet cell hyperplasia. Intestinal remodeling is a recognized sequela of helminth and protist infections which trigger a succinate-dependent and independent tuft cell circuit that induces the proliferation of specialized epithelial populations (i.e., tuft and goblet cells) and the induction of a type 2 immune response within the small intestines[24,27].

Activation of ILC2s is a critical component of this immune circuit since these cells express the IL-25R which makes them responsive to tuft cell-dependent secretion of IL-25[25]. This tuft cell-ILC2 circuit-triggered event is thought to be necessary to maintain an appropriate energy balance for the host to mount an anti-pathogen response[24]. Paneth cell metaplasia, however, is not a recognized feature of intestinal remodeling, nor does IL-25 signaling have any reported role in Paneth cell biology. Thus, we sought to define a mechanistic pathway that would link these seemingly disparate events. To that end, we identified IL-9 as the critical cytokine responsible for driving both PCM and intestinal remodeling events in CML mice (Fig. 9). IL-9 was specifically and differentially increased in the colon and ileum where PCM and Paneth cell hyperplasia were present. In addition, antibody-mediated blockade of this signaling pathway effectively inhibited the development of PCM as evidenced by a significant decrease in the absolute number of Paneth cells as well as mRNA levels of cryptdin 1 and sPLA2. Furthermore, inhibition of IL-9 also prevented the emergence of tuft and goblet cell hyperplasia, indicating that this cytokine mediated widespread small intestinal remodeling and altered the topography of specialized epithelial cell populations.

Metabolic profiling of the small intestines also revealed significant increases in histamine and its downstream metabolites, suggesting that CML had effects on other resident intestinal cell populations that are not commonly affected during small intestinal remodeling. Mast cells are the dominant histamine-producing tissue-resident cell population and exist as two subsets; a constitutive class that is distributed in the connective tissue, and an inducible class of mucosal mast cells that are interepithelial and reside in intestinal and respiratory mucosa[23]. Connective tissue mast cells in the mouse can be distinguished by the expression of mMCP 4,5,6, and 7 proteases, while mucosal mast cells express mMCP 1 and 2 which function as chymases. The fact that only chymase expression was increased in the small intestines of CML animals was indicative of mucosal mast cell hyperplasia, which was further verified by immunohistochemical staining, indicating that mast cells were the likely source of histamine in leukemic animals. With respect to IL-9, this cytokine has also been shown to enhance the survival and proliferation of mast cells[31] as these cells express the IL-9 receptor on the cell surface[47]. In fact, transgenic overexpression of IL-9 results in intestinal mastocytosis during nematode infections[48]. Notably, mast cells have also been implicated in the pathophysiology of inflammatory bowel diseases (IBD)[49,50] but have not been shown to have an etiological role in the emergence of PCM which is associated with IBD. In CML mice, we observed that mast cell hyperplasia was similarly regulated by IL-9, as anti-IL-9 antibody administration effectively abrogated hyperplasia in the ileum as confirmed by mast cell quantitation and gene expression of mast cell proteases. In contrast, IL-25 had no apparent role in the upstream regulation of IL-13, or the expression of genes associated with specialized epithelial cell populations in CML mice. Thus, IL-9 coordinately regulated both PCM and mast cell hyperplasia.

The unexpected effect of CML on intestinal remodeling and epithelial cell fate decisions led us to further examine the underlying

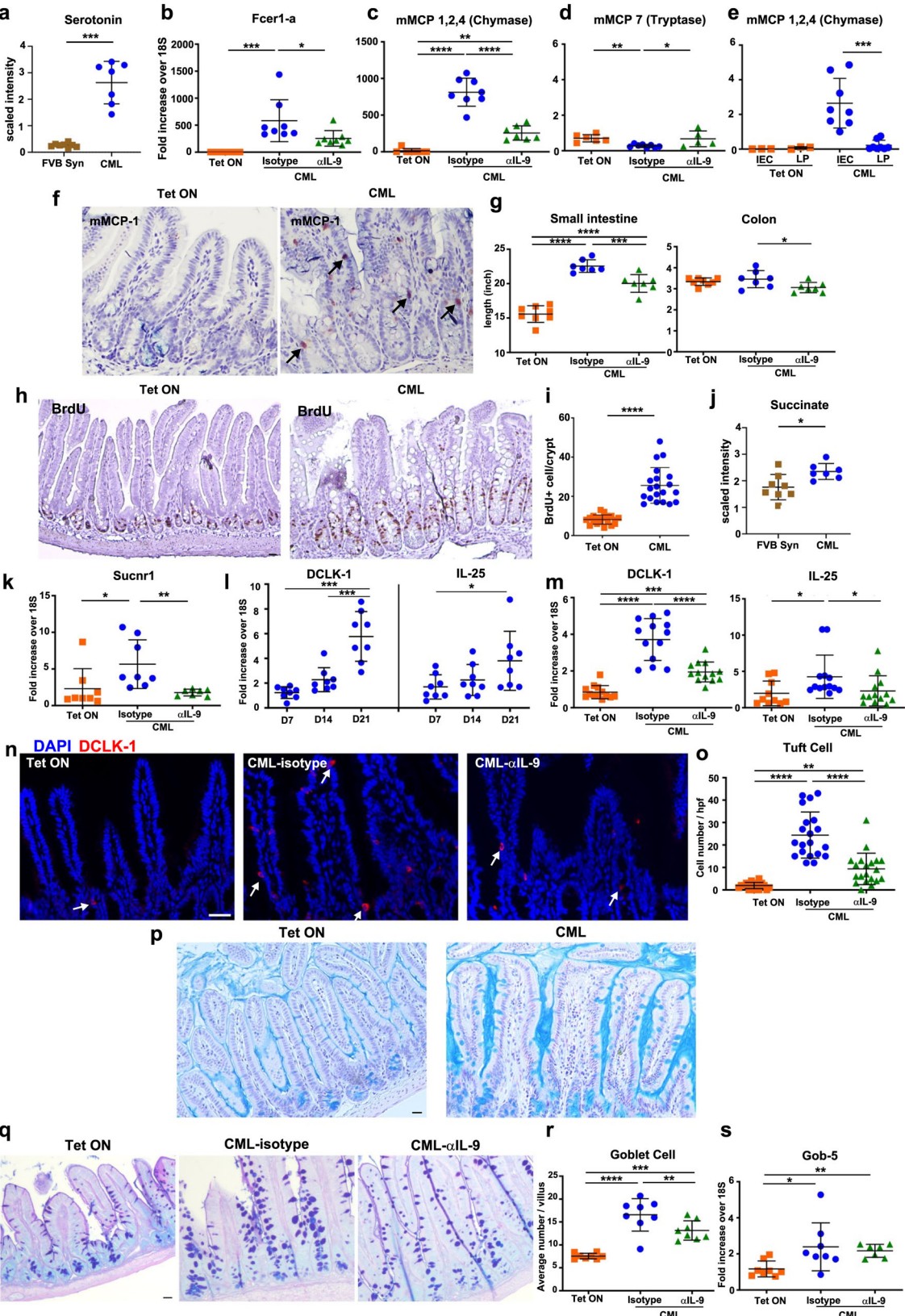

regulatory network downstream of IL-9 that helped to facilitate this remodeling event. Metabolic profiling studies revealed an increased level of succinate along with augmented expression of the succinate receptor in CML mice. The succinate receptor is known to be expressed at highest levels in tuft cells[27], and the increase in IL-25 and DCLK-1 expression along with marked small intestinal lengthening were all

indicative of tuft cell hyperplasia which was confirmed by immuno-fluorescence staining in the small intestines. Tuft cell hyperplasia has been shown to be dependent upon the production of IL-13 by type 2 ILCs which also induces goblet cell metaplasia[51]. Blockade of this pathway in CML mice resulted in a reduction in the number of tuft and goblet cells, as well as a significant decrease in Paneth cell-specific gene

**Fig. 4 | CML induces intestinal remodeling and hyperplasia of specialized epithelial cell populations. a** Scaled intensity of serotonin in the ileum of FVB recipients transplanted with BM from normal FVB (FVB Syn) or SCLtTa/bcr-abl (CML) donors($n$ = 7–8 mice/group). **b–d** FVB mice transplanted with BM from SCLtTa/bcr-abl mice and maintained on Tet or off Tet and treated with an isotype control or anti-IL-9 antibody. mRNA expression of Fcer1-a (**b**), mMCP 1,2,4 (**c**), and mMCP 7 (**d**) in the ileum. Results are from two experiments ($n$ = 6–8 mice/group). **e** mRNA expression of mMCP 1,2,4 from intestinal epithelial cells (IECs) and lamina propria cells (LP) in the colon 35 days post-transplantation. Data are from two experiments ($n$ = 3–8 mice/group). **f** Representative photomicrograph of mMCP 1,2,4 staining in the ileum. **g** Small intestine and colon length. Results are from two experiments ($n$ = 6–8 mice/group). **h** Immunohistochemical staining of the ileum depicting BrdU expression in the crypts of the intestinal epithelium of CML mice. **i** The number of BrdU+ cells in the crypts of mice. Data are from two experiments ($n$ = 20/group). **j** Scaled intensity of succinate in the ileum of recipients transplanted with BM from syngeneic or SCLtTa/bcr-abl donors ($n$ = 7–8 mice/ group). **k** mRNA expression of the succinate receptor (Sucnr1). ($n$ = 7–8 mice/group). **l** mRNA expression of DCLK-1 and IL-25 in the ileum. Results are from two experiments ($n$ = 8 mice/group). **m–o** FVB mice were transplanted with BM from SCLtTa/bcr-abl mice and maintained on or taken off Tet and treated with an isotype control or anti-IL-9 antibody. DCLK-1 and IL-25 gene expression. **m** Data are from three experiments ($n$ = 10–13 mice/group). Representative immunofluorescence image of DCLK-1+ cells (**n**) and absolute number of DCLK-1+ tuft cells per high power field (**o**) in the ileum. Results are from two experiments ($n$ = 20/group). **p** Mucus staining with Carnoy's fixative of the ileum. **q–s** Representative Alcian blue staining of goblet cells (**q**), absolute number of goblet cells (**r**), and mRNA expression of Gob-5 in the ileum (**s**). Data are from two experiments ($n$ = 8 mice/group). Analysis was performed 21–28 days post-transplantation in (**b–d, f, g, l, k, m–o**). Data are shown as mean ± SD. Statistics were performed using Welch's $t$ test or one-way ANOVA and Fisher's LSD test.: *$p$ < 0.05, **$p$ < 0.01, ***$p$ < 0.001, ****$p$ < 0.0001. Source data are provided as a Source Data file.

expression. However, whereas there was a reduction in crypt 1 gene expression in anti-IL-13 antibody-treated animals, there were no quantitative alterations in the absolute number of Paneth cells or other Paneth cell-specific peptides when compared to isotype antibody-treated control mice. Furthermore, IL-13 had no impact on the development of mast cell hyperplasia. Thus, IL-13, unlike IL-9, was more restricted to the regulation of tuft and goblet cell development in CML mice and had no effect on the emergence of PCM or mast cell hyperplasia.

Given the critical role of IL-9 in promoting PCM and intestinal remodeling, we sought to define the cell population that was responsible for the production of this cytokine in the GI tract. IL-9 has been shown to be produced by type 2 helper T cells[30,31], $T_H$17 cells[32,33], regulatory T cells[34], and innate lymphoid cells[35], and has effects on both hematopoietic and non-hematopoietic cell populations[32]. Therefore, we pursued an iterative approach to identify IL-9-expressing populations in the GI tract. Immunofluorescence staining demonstrated that there was a lack of co-localization of IL-9 and the mast cell marker mMCP-1, and IL-9 transcripts were found only in the lamina propria which was not the resident site for mucosal mast cells. In addition, IL-9 was not detectable in CD3+ T cells in the ileum or colon, and IL-9 transcripts were predominantly expressed in a non-T cell population in the mesenteric lymph nodes. Depletion of CD4+ T cells also had no effect on gene expression of IL-9, Paneth, tuft or mast cell markers, indicating that CD4+ T cells were not the source of IL-9. Conversely, we observed that IL-9 prominently co-localized with GATA3 and KLRG-1 which are markers for ILC2 cells[40]. In addition, there was significantly increased gene expression of IL-4, IL-5, ST2, and IL-17Rb in the small intestines which are characteristic of type 2 ILCs[37]. Thus, these data indicated that ILC2s were the dominant source of IL-9 in the GI tract.

To determine the upstream signaling events that led to the activation of ILC2s, we noted that CML induced an inflammatory environment within the GI tract characterized by increased expression of an array of proinflammatory cytokines (e.g., GM-CSF, TNF-α, IL-6, IFN-γ, and IL-22) as well as lipids that have been implicated in inflammation. The ability of CML to induce systemic inflammation has been reported in mice and newly diagnosed CML patients where increased levels of inflammatory cytokines and chemokines have been documented in the plasma and hematopoietic tissues[52–54]. This inflammatory environment was associated with increased cleaved caspase 3 expression in intestinal epithelial cells which is a marker for cell death. We observed that IL-33, which functions as an alarmin[42], along with its receptor ST2, were augmented in the ileum and colon of CML animals and localized to the epithelial cell layer. IL-33 in mice is secreted as a long form 266 amino acid protein that is initially localized in the nucleus of epithelial barrier tissues along with endothelial and epithelial cells in blood vessels[41]. Inflammatory proteases from neutrophils (i.e., elastase and cathepsin G)[43] and mast cells (i.e., chymase and tryptase)[44] can cleave this protein into smaller mature forms of this protein which have significantly enhanced (i.e., -10-30-fold) biological activity. In fact, we observed that there were smaller forms of IL-33 present within the GI tract along with increased protein expression of chymase which is stored as a secretory granule in mast cells. The association of mast cells in close proximity to ILC2s has been demonstrated in environmentally facing tissue sites[37]. We therefore speculate that the large number of mast cells in the mucosal layer was a source of chymase leading to the enzymatic cleavage of IL-33 into more potent forms that resulted in the activation of IL-9-producing ILC2s, thereby linking mast cells to the development of PCM (Fig. 9). This conclusion was further supported by the fact that blockade of IL-9 signaling completely abrogated expression of chymase and this was associated with disappearance of the short form of IL-33. Finally, inhibition of IL-33/ST2 signaling with an ST2 monoclonal antibody significantly reduced the expression of IL-9, along with other ILC2-derived cytokines, indicating that IL-33 was a critical upstream regulator of type 2 ILC-mediated IL-9 production. In murine allergy models, IL-2 produced by adaptive immune cells has also been shown to augment IL-9 production[35], so these results do not exclude other potentiating factors that might act in conjunction with IL-33 in CML mice.

In summary, our studies demonstrate that an extra intestinal disease can induce an inflammatory environment within the GI tract that leads to the production of IL-33 from stressed epithelial cell populations. Binding of IL-33 to ST2 induces type 2 ILC-mediated production of IL-9 which is the proximate cytokine responsible for the development of PCM as well as the emergence of extensive small intestinal remodeling characterized by hyperplasia of specialized epithelial cell populations. Thus, an IL-33-regulated ILC2/IL-9 immune circuit resident in the GI tract establishes a mechanistic pathway for the development of PCM which may have pathophysiological relevance for the emergence of this phenomenon arising in GI intrinsic diseases such as colorectal cancer and inflammatory bowel diseases, although this will require further study to validate this premise.

# Methods
## Ethical regulations
All experiments were carried out under protocols approved by the MCW Institutional Animals Care and Use Committee (IACUC).

## Mice
FVB/NJ (H-2$^q$) mice were bred in the Animal Resource Center at the Medical College of Wisconsin (MCW) or purchased from Jackson Laboratories (Bar Harbor, ME) (Stock number #001800). Transgenic mice in which a tetracycline-controlled transactivator was placed under the control of the murine stem cell leukemia gene 3' enhancer [SCLtTA mice (FVB/N background)] were crossbred to transgenic TRE-BCR-ABL mice (FVB/N background) which expressed the bcr/abl

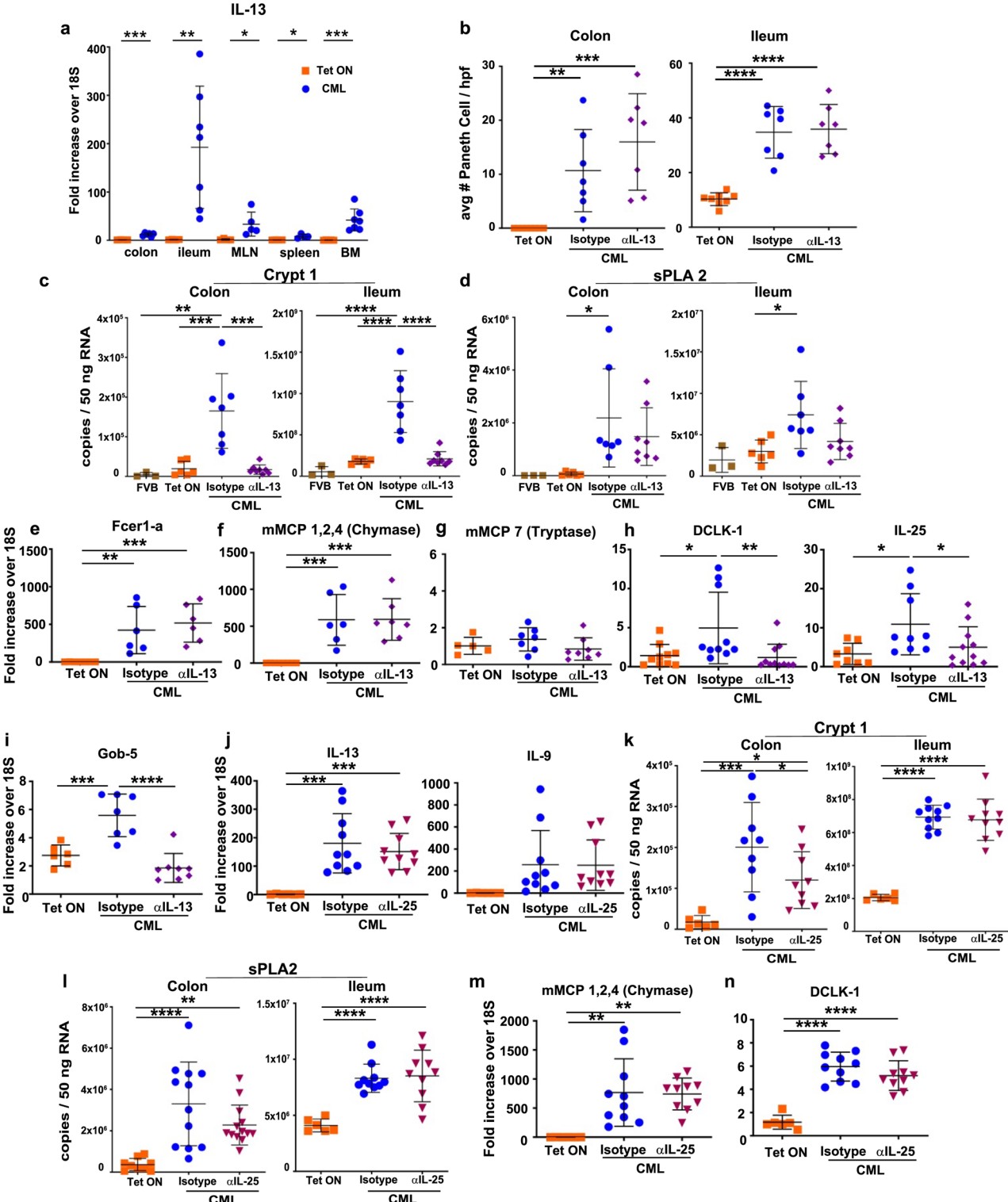

oncogene to create double transgenic SCLtTA x BCR-ABL mice (i.e., CML mice)[16]. Screening of SCL, bcr/abl, and double transgenic mice was conducted by PCR. Induction of bcr/abl expression by withdrawal of tetracycline (0.5 g/l) from the drinking water results in activation of the transgene and subsequent leukemogenesis, as previously described[16]. All animals were housed under specific pathogen free conditions in the American Association for Laboratory Animal Care (AALAC)-accredited Animal Resource Center of the Medical College of Wisconsin. Ambient temperature was 72 F with 30-33% humidity and

14/10 dark/light cycles. Mice received regular mouse chow and acidified tap water ad libitum.

**Tetracycline-dependent CML transplantation model**
Bone marrow (BM) was flushed from donor femurs and tibias with Dulbecco's modified media (DMEM, Gibco-BRL, Life Technologies, Grand Island, NY) and passed through sterile mesh filters to obtain single cell suspensions. A transplant model[17] was used in this study to achieve more uniform disease onset kinetics since CML arising de novo

**Fig. 5 | IL-13 regulates tuft and goblet cell hyperplasia but not Paneth cell metaplasia. a** Irradiated FVB mice were transplanted with BM from SCLtTa/bcr-abl animals. mRNA gene expression of IL-13 in the colon, ileum, mesenteric lymph nodes, spleen and bone marrow of animals that were maintained on or off Tet 21-22 days post-transplantation. Data are from two experiments ($n = 4-7$ mice/group). **b–i** FVB mice were transplanted with BM from SCLtTa/bcr-abl mice. Animals were maintained on or off Tet and then treated with an isotype control or anti-IL-13 antibody. **b–d** The average number of Paneth cells per 10 high power fields as determined by H&E staining (**b**), and mRNA expression of Crypt1 (**c**) and sPLA2 gene expression (**d**) in the colon and ileum. Results are from two experiments ($n = 3-8$ mice/group). **e–g** mRNA expression of Fcer1-a (**e**), mMCP 1,2,4 (**f**) and mMCP 7 (**g**) in the ileum. **h, i** mRNA expression of DCLK-1 and IL-25 (**h**) and Gob-5 (**i**) in the ileum. Analysis was performed 34–35 days post-transplantation. Data in (**e–i**) are from two–three experiments ($n = 5-11$ mice/group). **j–n** Lethally irradiated FVB mice were transplanted with BM from SCLtTa/bcr-abl mice. Animals were either maintained on or taken off Tet and then treated with an isotype control or anti-IL-25 antibody. **j** mRNA expression of IL-13 and IL-9 gene expression in the ileum. **k. l** mRNA expression of Crypt1 (**k**) and sPLA2 gene expression (**l**) in the colon and ileum. **m, n** mRNA expression of mMCP 1,2,4 (**m**) and DCLK-1 (**n**) in the ileum of mice. Analysis for these endpoints was performed 26 days post-transplantation. Data in (**j–n**) are from two–three experiments ($n = 6-13$ mice/group). Data are shown as mean ± SD. Statistics were performed using Welch's *t* test (pairwise comparisons), one-way ANOVA plus Fishers LSD test (three-way comparisons), and one-way ANOVA plus Tukey's correction (greater than three comparisons).: *$p < 0.05$, **$p < 0.01$,***$p < 0.001$, ****$p < 0.0001$. Source data are provided as a Source Data file.

in mice can have significant variability in disease onset[16]. To conduct these studies, bone marrow was taken from CML donors and passed through sterile filters to obtain single cell suspensions. Host FVB/N mice were conditioned with total body irradiation administered as a single exposure at a dose rate of 60 cGy using a Shepherd Mark I Cesium Irradiator (J.L. Shepherd and Associates, San Fernando, CA). Irradiated recipients (1100 cGy) received a single intravenous injection in the lateral tail vein of BM ($10 \times 10^6$) in a total volume of 0.4 ml. Donor and recipient mice were all 6-12 weeks of age and sex matched in all transplant experiments. Animals in experimental/control groups were not co-housed. When applicable, mice were euthanized by cervical dislocation per IACUC approved guidelines.

### Reagents

Anti-IL-9 (MM9C1) is a previously described mouse IgG2a antibody (35) that was administered intraperitoneally at a dose of 200 µg three times per week. Mouse IgG2a (C140SF9) was used as an isotype control and administered at the same dose and schedule. Both anti-IL-9 and the isotype antibody were purified from the culture supernatant of hybridoma cells using the Proteus protein purification spin kit (Bio-Rad, Hercules, CA). Anti-IL-13 is an IgG1 antibody[55] that was administered at a dose of 10 mg/kg three times a week by intraperitoneal injection. Mouse IgG1 was used as an isotype control antibody (BioX-Cell, West Lebanon, NH). Anti-CD4 (clone GK1.5, rat IgG2b) (BioXCell) and rat IgG isotype antibody (Jackson ImmunoResearch, West Grove, PA) were administered at a dose of 250 µg twice per week. Anti-ST2 (R&D Systems, Minneapolis, MN, clone 245707, rat IgG2b) and rat IgG isotype antibody (Jackson ImmunoResearch) were administered at a dose of 50 or 100 µg three times per week starting at day 14 post transplantation. Anti-IL-5 (BioXCell, clone TRFK5) and rat IgG isotype antibody (Jackson ImmunoResearch) were administered at a dose of 500 µg twice per week. Anti-IL-25 (clone 2C3.2)[56] was purified from the culture supernatant of hybridoma cells using the Proteus protein purification spin kit. Anti-IL-25 and rat IgG isotype (Jackson ImmunoResearch) were administered at a dose of 500 µg three times per week.

### Blood collection and complete blood count (CBC)

Blood samples were obtained by venipuncture of the facial (submandibular) vein to monitor the progression of CML. Complete blood counts were performed using a scil Vet ABC ™ Hematology Analyzer (Scil Animal Care Company, Gurnee, Ill).

### Phloxine-Tartrazine staining for Paneth cells

Formalin-fixed paraffin embedded (FFPE) tissue samples were cut into 5-µm sections and dewaxed with xylene and ethanol. The rehydrated sections were stained with Harris modified hematoxylin solution (Thermo Fisher Scientific, Carlsbad, CA) for visualization of nuclei in tissue sections. The sections were then immediately stained with 0.5% phloxine (Sigma, St. Louis, MO) in 0.5% $CaCl_2$ solution, and differentiated with a saturated tartrazine (Sigma) solution. Paneth cell

granules were identified under light microscopy as red-appearing granules.

### Histological analysis of specialized intestinal epithelial cells

Distal colons and small intestines were fixed in 10% neutral-buffered formalin and embedded in paraffin. Formalin-fixed paraffin embedded (FFPE) tissue samples were then cut into 5-µm sections and stained with hematoxylin and eosin. Paneth cells were quantitated by selecting 10 randomly picked high power fields and averaging the number of Paneth cells per 10 high-power fields for each experimental group. The average number of Goblet cells in 10 randomly selected villi in the ileum per sample were quantitated. Tuft cells were enumerated by randomly selecting 5 high-power fields per sample with each data point representing the cell count per one high power field. The extent of underlying inflammation in the colons of mice that were maintained on or off Tet was determined using a semiquantitative scoring system that was previously developed for the evaluation of GVHD of the colon and included an assessment of crypt abscesses, lamina propria inflammation, epithelial cell ulceration, and mucin depletion[37,38]. All slides were coded and read in a blinded fashion. Light microscopy images were visualized with a Nikon ECLIPSE E400 microscope (Tokyo, Japan). Image acquisition was performed with a Nikon DS-Fi3 colorimetric digital camera and software package.

### Staining of GI tract mucus

To preserve and firm the mucus layer, harvested ileum tissue was placed in Carnoy's solution (Electron Microscopy Sciences, Hatfield, PA) for 4 h. After fixation, tissues were dehydrated through graded ethanol, cleared with xylene, and paraffin infiltrated using an automated tissue processor (Sakura, Tissue-TEK VIP5). After orientation tissue samples were embedded into paraffin blocks, sectioned at 4um and mounted onto coated slides. Alcian blue-periodic acid Schiff (AB-PAS, pH2.5, Thermo Fisher Scientific) staining was performed to visualize mucus in the ileum.

### Immunohistochemistry

Paraffin-embedded tissue samples were de-waxed with CitriSolv (DeconLabs, King of Prussia, PA) and rehydrated before performing immunohistochemistry staining using Utra-Sensitive ABC Peroxidase Staining Kit (Thermo Fisher Scientific). Samples were immersed in EDTA Antigen Retrieval Buffer (pH 8.5, Sigma) on a hot plate and simmered for 25 min. Endogenous peroxidase activity was quenched with Peroxidase Suppressor Buffer (Thermo Fisher Scientific) for 30 min at room temperature. Samples were blocked and stained with primary antibody overnight at 4 °C. After fully washing with BupH Tris Buffered Saline (TBS, Thermo Fisher Scientific), samples were incubated with biotinylated secondary antibody and developed with HRP and substrate using a DAB substrate kit (Thermo Fisher Scientific). Primary and secondary antibodies used in this study are listed in Supplementary Table 1. Images were visualized with a Nikon ECLIPSE

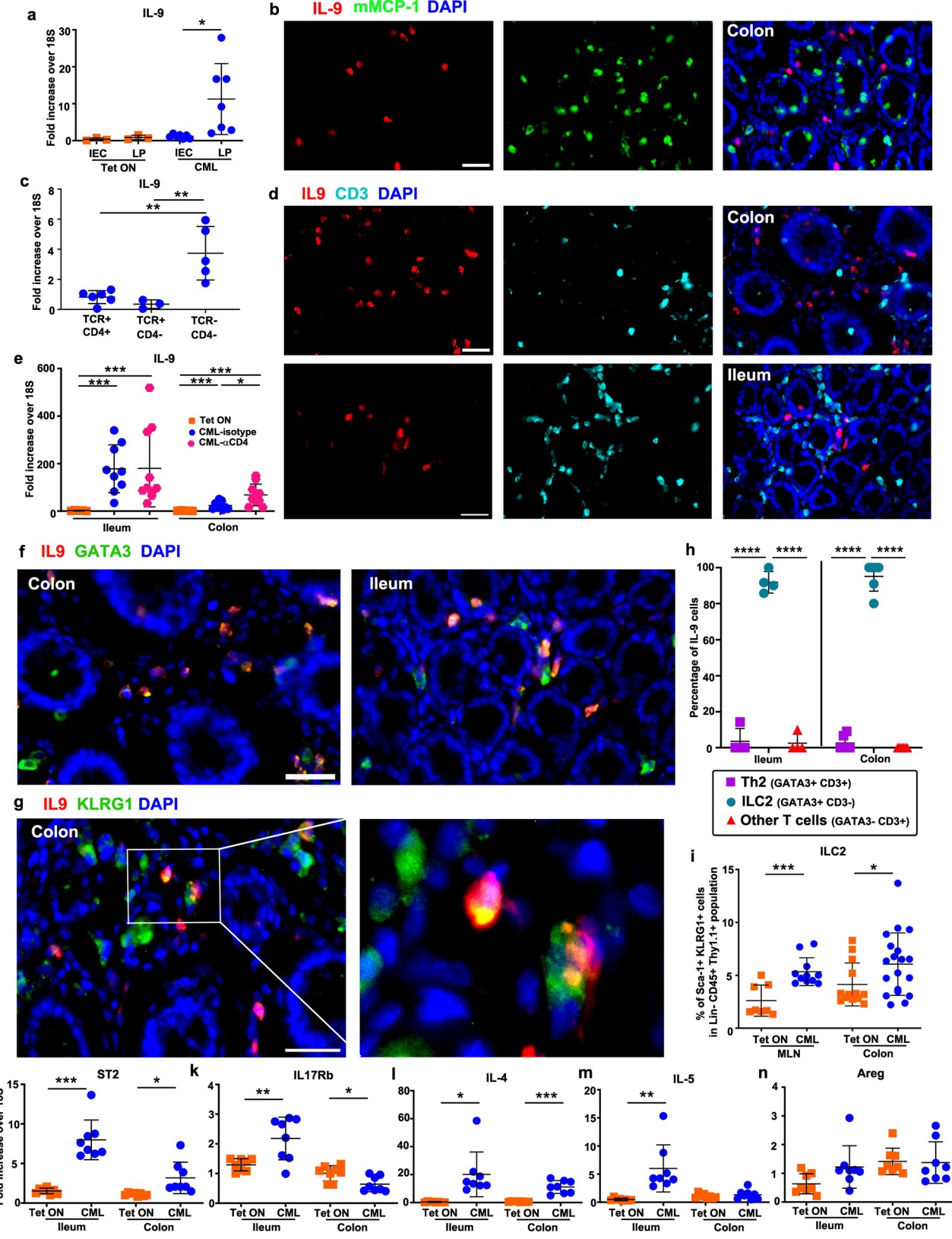

E400 microscope. Image acquisition was performed with a Nikon DS-Fi3 colorimetric camera.

## Immunofluorescence

Distal colons and small intestines were fixed in 4% formaldehyde and embedded in O.C.T. compound (Sakura, Torrance, CA) for cryosections. Tissue samples were then cut into 12-micron sections for immunofluorescence staining using a Leica CM1520 cryostat (Leica Biosystems, Deer Park, IL). Samples were first fixed in methanol at −20 °C then treated with Antigen Retrieval Buffer (pH 6, Sigma). 0.3% Triton X-100 (Sigma) was used to permeabilize the sample sections. Samples were blocked and incubated with primary antibody overnight

**Fig. 6 | Type 2 ILCs are the primary source of IL-9. a** Irradiated FVB mice were transplanted with BM from SCLtTa/bcr-abl animals. Mice were maintained on or off Tet. mRNA expression of IL-9 from intestinal epithelial (IECs) and lamina propria cells (LP) 28–35 days post-transplantation. Data are from two experiments (*n* = 3–7 mice/group). **b** Immunofluorescence staining of the colon showing isolated and merged expression of mMCP-1 and IL-9 in CML mice. **c** IL-9 gene expression in sorted TCR$^+$ CD4$^+$, TCR$^+$ CD4$^-$ and TCR$^-$ CD4$^-$ populations from the mesenteric lymph nodes of CML mice 4-5 weeks post-transplantation. Results are from two experiments (*n* = 3–6 mice/group). (**d**). Immunofluorescence staining of the ileum and colon showing isolated and merged expression of CD3 and IL-9. **e** FVB mice were transplanted with BM from SCLtTa/bcr-abl mice. Animals were maintained on or off Tet and treated with an isotype control or anti-CD4 antibody for 4 weeks. IL-9 gene expression in the ileum and colon. Data are from two experiments (*n* = 5–10/ group). **f–n** FVB mice were transplanted with BM cells from SCLtTa/bcr-abl animals and maintained off Tet. **f** Immunofluorescence staining in the ileum and colon showing isolated and merged expression of GATA3 and IL-9 in CML mice. **g** Immunofluorescence staining in the colon showing isolated and merged expression of KLRG1 and IL-9. **h** Percentage of IL-9-expressing cells that co-expressed CD3 or GATA3 in the ileum or colon of CML mice as determined by immunofluorescence staining. Results are from 4–6 mice/group. **i** Frequency of ILC2s in the mLN and colon 4 weeks post-transplantation. Data are from three experiments (8–19 mice/group). **j–n** mRNA expression of ST2 (*Il1rl1*), IL-17RB, IL-4, IL-5, and amphiregulin (Areg) in the ileum and colon 21 days post-transplantation in FVB mice transplanted with SCLtTa/bcr-abl BM and then maintained on or off Tet. Results are from two experiments (*n* = 8 mice/group). Statistics were performed using Welch's *t* test (**a**, **h–n**) or one-way ANOVA and Fisher's LSD test (**c**, **e**). **p* ≤ 0.05, ***p* < 0.01, ****p* < 0.001. Scale bar = 30 μm for all panels. Data are shown as mean ± SD. Source data are provided as a Source Data file.

and incubated with fluorochrome-conjugated secondary antibody on the next day. Primary and secondary antibodies used in this study are listed in Supplementary Table 1. Images were visualized with a Nikon ECLIPSE TE2000-U microscope. Image acquisition was performed with NES elements software package.

### Quantification of IL-9-expressing cells in the GI tract

Immunofluorescence-stained tissues were employed to quantify the percentage of IL-9 expressing cells that were GATA3$^+$ CD3$^+$, GATA3$^+$ CD3$^-$, GATA3$^-$ CD3$^+$ in cross-sections of colons and ileums from CML mice. The brightness/contrast of the acquired digital images was applied equally across the entire image and equally to control images and analyzed using Image J/FIJI software (National Institute of Health, Bethesda, MD). Automated protocols for signal intensity (SI) and masks were created using ~2 SD SI threshold from the Mean Fluorescence Intensity (MFI) with background SI subtracted. Image J/FIJI software was also used to determine ideal threshold to define colocalization between channels. Five–eight fields of view (FOV) per organ per mouse in analogous regions between samples were analyzed to accurately represent the whole tissue.

### Western blot analysis

Tissues harvested from mice were homogenized using a homogenizer (PowerGen 125, Thermo Fisher Scientific) in RIPA buffer (Thermo Fisher Scientific, Cat. 89900) with the appropriate dilution of protease and phosphatase inhibitors (Thermo Fisher Scientific, Cat. A32959). The homogenized tissue was then stored at −80 °C. Protein levels were normalized using a BCA Assay. Primary antibodies used for western blot were used as follows: rat anti-IL-33-unconjugated (1:1000; R&D Systems MAB3626), rat anti-MCP1/Mcpt1 (1:1000; R&D Systems MAB5416), rabbit anti-Cathepsin G (1:3000; ThermoFisher; PA5-89049), rabbit anti-Neutrophil Elastase (1:3000; Cell Signaling; E8U3X), rabbit anti-cleaved caspase 3 [ASP175], (1:3000; Cell Signaling; 9664 S), and mouse anti-ß-actin (1:5000; Cell Signaling; E4D9Z). The appropriate HRP-conjugated secondary antibodies were used at 1:5000 (R&D Systems). For a detailed list of reagents, refer to Supplementary Table 1. Relative expression of proteins was assessed using BioRad chemiluminescence imaging system.

### Electron microscopy

Paraffin-embedded colonic samples were dewaxed in 3 × 24 h changes of xylene and rehydrated through decreasing concentrations of ethanol in distilled water. After rehydration, the sample was washed in 0.1 M sodium cacodylate buffer (2 × 10 min) then fixed in 2% glutaraldehyde in 0.1 M cacodylate buffer for one hour. After three 10 min washes in buffer, the sample was post-fixed in 1% aqueous osmium tetroxide for one hour on ice, washed in distilled water, dehydrated through graded methanol and embedded in epoxy resin (EMBed812). Ultrathin sections (60 nm) were stained with uranyl acetate and lead citrate and viewed in a JEOL2100 transmission electron microscope

(Japanese Electron Optics LtD, Tokyo, Japan) and images recorded using a Gatan Ultrascan CCD camera (Pleasanton, CA).

### Splenocyte Isolation

Splenocytes were obtained by grinding the tissues though a mesh screen with a syringe plunger. Red blood cells in the cell suspension were lysed with Ammonium Chloride Tris (ACT) lysis buffer, prepared with ammonium chloride solution and Tris-HCl solution, pH 7.2. The cell suspension was subsequently filtered through a cell strainer and prepared for further analysis.

### Isolation of intestinal epithelial cells and lymphocytes in the colon

Epithelial cells were isolated from colon samples in the pre-digestion buffer using the Lamina Propria Dissociation Kit (Miltenyi Biotec, Auburn, CA) according to the manufacturer's instructions. To isolate lymphocytes from the lamina propria, colon samples were first washed in DMEM medium with DTT and EDTA. Samples were then digested with 10 μg/ml liberase TL (Roche, Basel, Switzerland) and 0.05% DNase (QIAGEN, Hilden, Germany), and processed using the gentleMACS Dissociator (Miltenyi). The resulting cell suspension was then layered on a 44%/67% Percoll gradient (Sigma).

### Evaluation of epithelial cell proliferation

BrdU (Bromodeoxyuridine, Sigma-Aldrich) was administered in vivo to label proliferating epithelial cells. Mice were peritoneally injected with BrdU solution at 100 mg/kg body weight. Ileum tissue was harvested 5 h after BrdU treatment. Harvested tissues were fixed in 10% neutral-buffered formalin and embedded in paraffin. Immunohistochemistry staining of BrdU was performed using the protocol described above. The antibody used in this study is listed in Supplementary Table 1.

### Flow Cytometry

Cells were re-suspended in Fluorescence Activated Cell Sorting buffer (FACS buffer, 2% FBS in PBS) and pre-stained with LIVE/DEAD Fixable Aqua (Thermo Fisher Scientific) according to the manufacturer's instructions to exclude dead cells. Cells were then labeled with fluorescently conjugated monoclonal antibodies as listed in Supplementary Table 1. Flow cytometry was used to identify CD4$^+$ T cells (CD4$^+$ TCRβ$^+$), granulocytes (Gr1$^+$ CD11b$^+$), ILC2 (CD45$^+$ Lin$^-$ Thy1$^+$ KLRG1$^+$ Sca-1$^+$), macrophages (CD45$^+$ TCR$^-$ CD64$^+$ CD11b$^+$), neutrophils (CD45$^+$ TCR$^-$ CD64$^-$ Ly6G$^+$ CD11b$^+$) and dendritic cells (CD45$^+$ TCR$^-$ CD64$^-$ Ly6G$^-$ CD11c$^+$ IAb$^+$) with surface markers. Cells were analyzed on an LSR-II or Fortessa X-20 flow cytometer with FACSDiva software (BD). Data were analyzed using FlowJo software (BD) (Version 9).The gating strategy to identify ILC2 cells is depicted in Supplementary Fig. 9.

### Cell Sorting

Mesenteric lymph nodes (mLN) cells from mice were sorted on a FACSAria II cell sorter (BD) using a 100-micron nozzle. Prior to cell

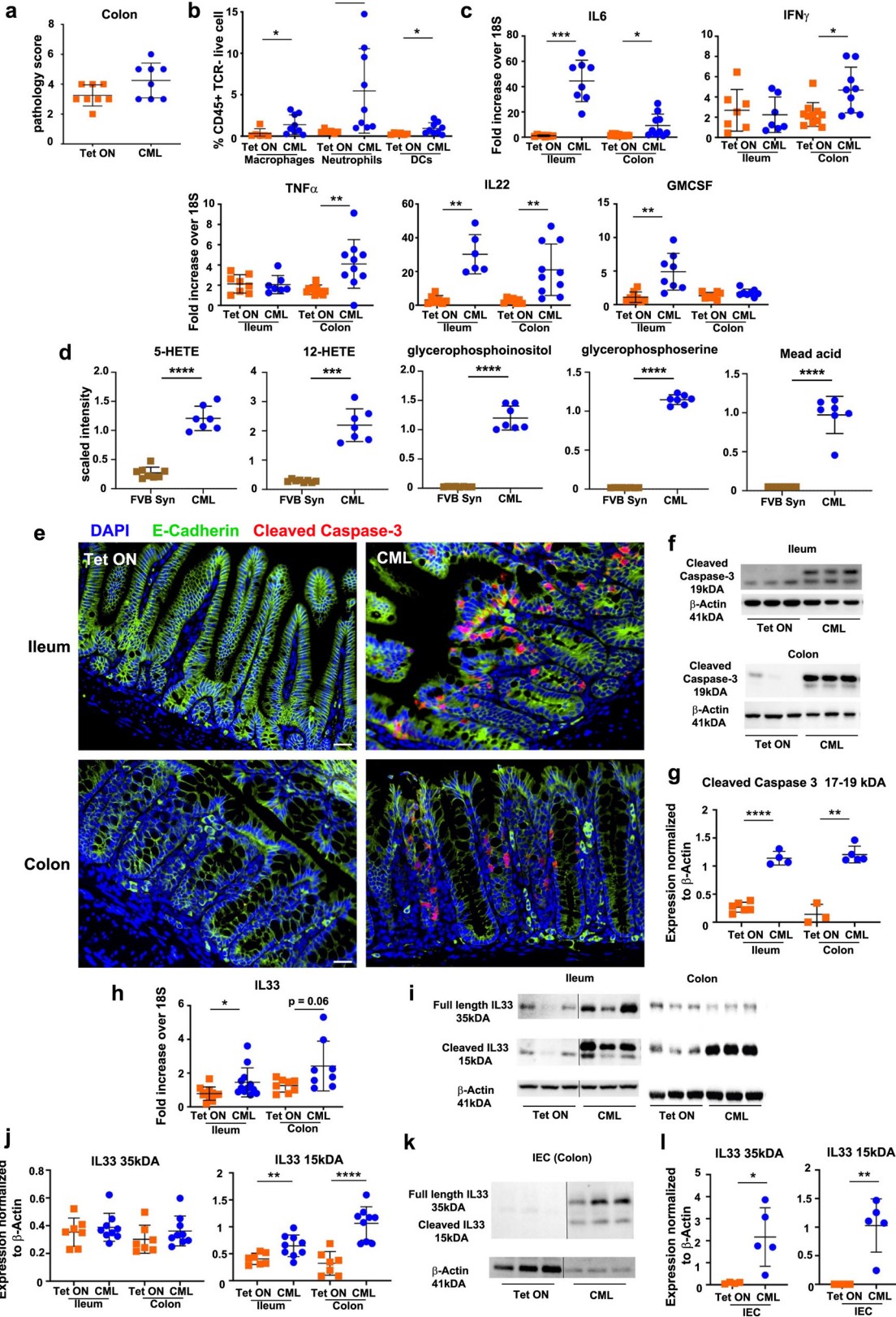

sorting, mLN cells were first enriched for CD4$^+$ T cells using magnetic purification (Thermo Fisher). Cells were then flow sorted into three populations (i.e., CD4$^-$ TCRβ$^-$, CD4$^+$ TCRβ$^+$ and CD4$^-$ TCRβ$^+$ cells). The quality of the sorted product was verified after each sort run. The gating strategy is depicted in Supplementary Fig. 9.

## RNA extraction and precipitation

Tissues harvested from mice were homogenized using a homogenizer (PowerGen 125, Thermo Fisher Scientific) in TRIzol RNA Isolation Reagent (Thermo Fisher Scientific). The homogenizer was previously cleaned with diethyl pyrocarbonate (DEPC) water to inactive RNase.

**Fig. 7 | CML induces inflammation in the GI tract and production of intestinal epithelial-derived IL-33. a–c** Lethally irradiated FVB mice were transplanted with SCLtTa/bcr-abl BM and maintained on or off Tet. **a** Pathological scores of colon samples from animals 34–39 days post-transplantation. **b** The percentage of macrophages, neutrophils and dendritic cells in the colon 30 days post-transplantation. (**c**). IL-6, IFN-γ, TNF-α, IL-22, and GM-CSF mRNA expression in the colon of animals 21 days post-transplantation. **d** FVB mice were transplanted with BM cells from SCLtTa / bcr-abl animals (CML) or normal FVB animals (FVB SYN). Scaled intensity of 5-HETE, 12-HETE, mead acid, glycerophosphoinositol and glycerophosphoserine in the ileum. **e–j** FVB mice were transplanted with SCLtTa/bcr-abl BM and maintained on or off Tet. **e** Immunofluorescence staining in the ileum and colon showing E-cadherin (green), cleaved caspase 3 (red) and DAPI (blue) expression in intestinal epithelial cells. Scale bar = 30 µm. **f, g** Immunoblot of cleaved caspase 3 expression in IECs in the ileum and colon (**f**) and summary data (**g**). **h** mRNA gene expression of IL-33 in the colon and ileum. **i, j** Immunoblot of IL-33 in the ileum and colon of CML mice demonstrating long (35 kDa) and short (15 kDa) forms (**i**) and summary data (**j**). **k, l** Immunoblot of IL-33 in intestinal epithelial cells (IECs) in the colon of CML mice (**k**) and summary data (**l**). Vertical lines on western blots denote non-contiguous gel lanes. Data are shown as mean ± SD. Statistics were performed using the Welch's $t$ test (pairwise comparisons). *$p \leq 0.05$, **$p < 0.01$, ***$p < 0.001$. Source data are provided as a Source Data file.

Chloroform and isopropyl alcohol were added to the TRIzol tissue lysate to precipitate RNA. For isolated cells, RNA was fixed by treating with RNAlater (Thermo Fisher Scientific) at 4 °C prior to processing extracted RNA. RNA was extracted using the RNeasy Mini Kit (QIAGEN).

## Quantitative real-time PCR for anti-microbial peptides and cytokines

To evaluate the expression of mouse AMPs, an absolute quantification method using standard curves was employed. Plasmid DNA of sPLA2 and Crypt1 was prepared using the QIAprep Spin Miniprep Kit (Qiagen) and used to generate standard curves[57]. RNA was isolated from homogenized colonic and ileal tissues in trizol. cDNA was then reverse-transcribed using the QIAGEN RT kit (QIAGEN) according to the manufacturer's instructions and subjected to RT-PCR using the QIAGEN QuantiTect SYBR Green PCR Master Mix in triplicate. Samples were run in a CFX C1000 Real-time Thermal Cycler (Bio-Rad, Hercules, CA). Standard curves were generated using Bio-Rad software, and the numbers of genomes per sample were extrapolated from the Ct values. Data are presented as copy number of AMPs genomes per 50 ng of mouse RNA.

To determine gene expression values of cytokines, RNA and cDNA were prepared and run as described above with the following modifications. An 18S reference gene was amplified using the QuantiTect Primer Assay Kit (Qiagen). The primers were purchased from Integrated DNA Technologies (Coralville, IA) and are listed in Supplementary Table 2. Specificity for all q-PCR reactions was verified by melting curve analysis. Cytokine values in different cohorts were normalized to an internal control 18S. To calculate fold-change in gene expression, the average ΔΔCt values from triplicate wells were combined from separate experiments.

## Metabolic profiling

Metabolic profiling of ileal samples from CML and non-leukemic control animals was conducted by Metabolon (Morrisville, NC). Samples were prepared using the automated MicroLab STAR® (Hamilton Company, Reno, NV). Several recovery standards were added prior to the first step in the extraction process for QC purposes. To remove protein, dissociate small molecules bound to protein or trapped in the precipitated protein matrix, and to recover chemically diverse metabolites, proteins were precipitated with methanol under vigorous shaking for 2 min (Glen Mills GenoGrinder 2000, Clifton, NJ) followed by centrifugation. The resulting extract was divided into five fractions: two for analysis by two separate reverse phase (RP)/UPLC-MS/MS methods with positive ion mode electrospray ionization (ESI), one for analysis by RP/UPLC-MS/MS with negative ion mode ESI, one for analysis by HILIC/UPLC-MS/MS with negative ion mode ESI, and one sample was reserved for backup. Samples were placed briefly on a TurboVap® (Zymark, Hopkinton, MA) to remove the organic solvent. The sample extracts were stored overnight under nitrogen before preparation for analysis.

Several types of controls were analyzed in concert with the experimental samples: a pooled matrix sample generated by taking a small volume of each experimental sample (or alternatively, use of a pool of well-characterized human plasma) served as a technical replicate throughout the data set; extracted water samples served as process blanks; and a cocktail of quality control (QC) standards that were carefully chosen not to interfere with the measurement of endogenous compounds were spiked into every analyzed sample, allowed instrument performance monitoring and aided chromatographic alignment. Instrument variability was determined by calculating the median relative standard deviation (RSD) for the standards that were added to each sample prior to injection into the mass spectrometers. Overall process variability was determined by calculating the median RSD for all endogenous metabolites (i.e., non-instrument standards) present in 100% of the pooled matrix samples. Experimental samples were randomized across the platform run with QC samples spaced evenly among the injections.

All methods utilized a Waters ACQUITY ultra-performance liquid chromatography (UPLC) and a Thermo Scientific Q-Exactive high resolution/accurate mass spectrometer interfaced with a heated electrospray ionization (HESI-II) source and Orbitrap mass analyzer operated at 35,000 mass resolution. The sample extract was dried then reconstituted in solvents compatible to each of the four methods. Each reconstitution solvent contained a series of standards at fixed concentrations to ensure injection and chromatographic consistency. One aliquot was analyzed using acidic positive ion conditions, chromatographically optimized for more hydrophilic compounds. In this method, the extract was gradient eluted from a C18 column (Waters UPLC BEH C18-2.1 × 100 mm, 1.7 µm) using water and methanol, containing 0.05% perfluoropentanoic acid (PFPA) and 0.1% formic acid (FA). Another aliquot was also analyzed using acidic positive ion conditions; however, it was chromatographically optimized for more hydrophobic compounds. In this method, the extract was gradient eluted from the same afore mentioned C18 column using methanol, acetonitrile, water, 0.05% PFPA and 0.01% FA and was operated at an overall higher organic content. Another aliquot was analyzed using basic negative ion optimized conditions using a separate dedicated C18 column. The basic extracts were gradient eluted from the column using methanol and water, however with 6.5 mM Ammonium Bicarbonate at pH 8. The fourth aliquot was analyzed via negative ionization following elution from a HILIC column (Waters UPLC BEH Amide 2.1 × 150 mm, 1.7 µm) using a gradient consisting of water and acetonitrile with 10 mM Ammonium Formate, pH 10.8. The MS analysis alternated between MS and data-dependent MS$^n$ scans using dynamic exclusion. The scan range varied slighted between methods but covered 70-1000 m/z. Raw data files are archived and extracted as described below.

Raw data was extracted, peak-identified and QC processed using Metabolon's hardware and software. These systems are built on a web-service platform utilizing Microsoft's .NET technologies, which run on high-performance application servers and fiber-channel storage arrays in clusters to provide active failover and load-balancing. Compounds were identified by comparison to library entries of purified standards or recurrent unknown entities. Metabolon maintains a library based on authenticated standards

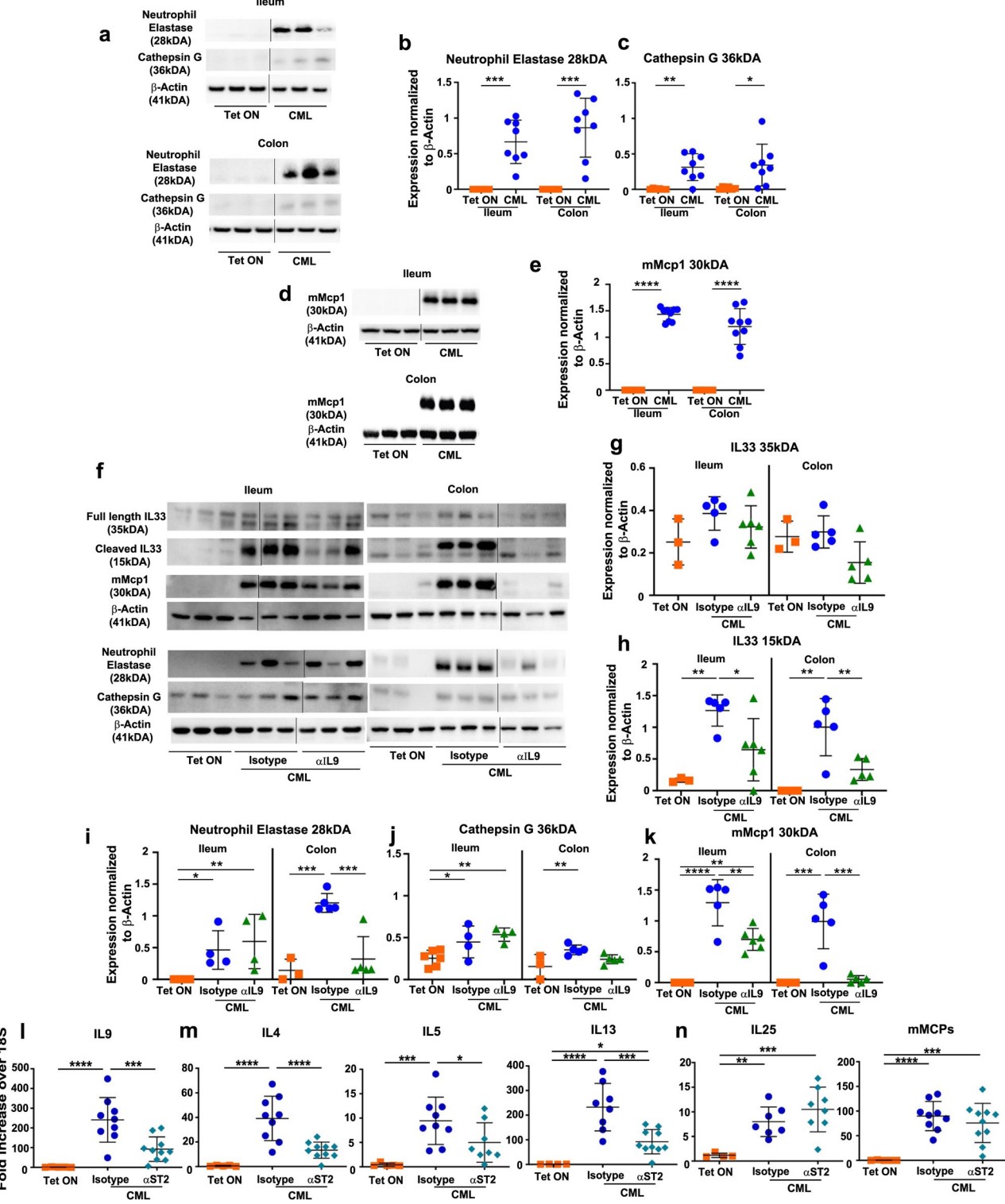

**Fig. 8 | IL-33 regulates the expression of ILC2-derived cytokines in the GI tract of CML mice. a–e** Irradiated FVB mice were transplanted with SCLtTa/bcr-abl BM and maintained on or off Tet. **a. b** Immunoblots of neutrophil elastase and cathepsin G in the ileum and colon of CML animals (panel a) and summary data (**b, c**). **d, e** Immunoblot of mMCP-1 in the ileum and colon of CML animals (**d**) and summary data (**e**). Results in (**b, c, e**) are from two experiments (*n* = 6–9 mice/group). **f–k** Irradiated FVB mice were transplanted with BM from SCLtTa/bcr-abl mice and then maintained on Tet or off Tet and treated with an isotype control or anti-IL-9 antibody. Immunoblot of IL-33 (35 kDa and 15 kDa), mMCP-1 (chymase), neutrophil elastase and cathepsin G in the ileum and colon (**f**), and summary data (**g–k**). Data in

(**g** and **h–k**) are from two experiments (*n* = 3–6 mice/group). **l–n** Irradiated FVB mice were transplanted with SCLtTa/bcr-abl BM and maintained on Tet or taken off Tet and treated with an isotype control or anti-ST2 antibody for two weeks beginning on day 14 post-transplantation. mRNA gene expression of IL-9 (**l**), IL-4, IL-5, IL-13 (**m**), IL-25 and mMCP 1,2,4 (**n**) in the ileum. Results in (**l–n**) are from two experiments (*n* = 4–10 mice/group). Vertical lines on western blots denote non-contiguous gel lanes. Data are shown as mean ± SD. Statistics were performed using Welch's *t* test (pairwise comparisons) or one-way ANOVA and Fishers LSD test (three-way comparisons). *$p \le 0.05$, **$p < 0.01$, ***$p < 0.001$. Source data are provided as a Source Data file.

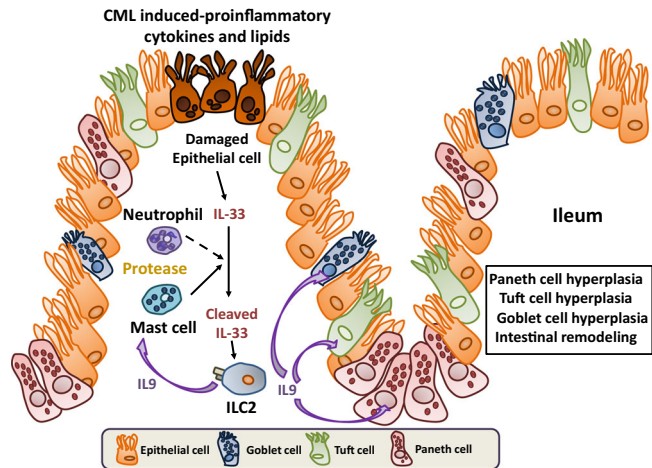

**Fig. 9 | Proposed model for the development of PCM and hyperplasia of specialized epithelial cell populations by an IL-33-induced ILC2/IL-9 immune circuit.** Epithelial-derived IL-33 is cleaved by mast cell chymase and, to a lesser extent, cathepsin G and neutrophil elastase, resulting in the production of the cleaved, more potent 15 kDa form of IL-33. Cleaved IL-33 binds to ST2 expressed on the surface of type 2 ILCs in the GI tract leading to the production of IL-9 which induces PCM and Paneth cell hyperplasia in the colon and ileum, respectively. IL-9 also induces hyperplasia of specialized epithelial cell populations (i.e., tuft and goblet cell hyperplasia) along with mucosal mast cell hyperplasia in the ileum.

that contains the retention time/index (RI), mass to charge ratio (*m/z*), and chromatographic data (including MS/MS spectral data) on all molecules present in the library. Furthermore, biochemical identifications are based on three criteria: retention index within a narrow RI window of the proposed identification, accurate mass match to the library +/− 10 ppm, and the MS/MS forward and reverse scores between the experimental data and authentic standards. The MS/MS scores are based on a comparison of the ions present in the experimental spectrum to the ions present in the library spectrum. While there may be similarities between these molecules based on one of these factors, the use of all three data points can be utilized to distinguish and differentiate biochemicals. More than 3300 commercially available purified standard compounds have been acquired and registered into LIMS for analysis on all platforms for determination of their analytical characteristics. Additional mass spectral entries have been created for structurally unnamed biochemicals, which have been identified by virtue of their recurrent nature (both chromatographic and mass spectral). These compounds have the potential to be identified by future acquisition of a matching purified standard or by classical structural analysis.

A variety of curation procedures were carried out to ensure that a high-quality data set was made available for statistical analysis and data interpretation. The quality control and curation processes were designed to ensure accurate and consistent identification of true chemical entities and to remove those representing system artifacts, misassignments and background noise. Metabolon data analysts used proprietary visualization and interpretation software to confirm the consistency of peak identification among the various samples. Library matches for each compound were checked for each sample and corrected if necessary.

### Statistics and reproducibility
Welch's *t* test was used to determine significant differences in pair wise group comparisons. For three-way comparisons within groups, results were compared using the one-way ANOVA plus Fishers LSD test. Experiments in which there were more than three groups were analyzed using the one-way ANOVA plus Tukey's correction. Statistical analysis was done using Prism software (GraphPad). For comparison of weight curves, a mixed effect model was fitted to the observations with fixed day and group effects with their interaction to model the mean, and a random subject-specific intercept and first-order continuous auto-regressive correlation to model the dependence structure. Based on the fitted model the change from day 0 to each following measurement day was compared between the three treatment groups using appropriate contrasts. Holm's method was used for multiple testing adjustment. All the analyses were performed in R version 3.4.1 (2017-06-30), using the nlme 3.1.131 package for fitting the mixed model, and multcomp 1.4.6 for multiple testing. No statistical method was used to predetermine the sample size. No data were excluded from the analysis. The experiments were not randomized, and investigators were not blinded to allocations during experiments or outcome assessments. A *p* value of ≤0.05 was deemed to be significant in all experiments.

### Reporting summary
Further information on research design is available in the Nature Portfolio Reporting Summary linked to this article.

## Data availability
The authors declare that all data pertaining to the current study are available within the article and Supplementary Information or available from the corresponding author upon request. Source data are provided with this paper.

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

## Acknowledgements

This research was supported by grants from the National Institutes of Health (HL R01 064603 and HL R01 126166) (W.R.D) and GM R01 099526 (N.S.). We thank Drs. Kate Dixon and Roger Palframan from UCB Pharma, Cambridge, MA for the provision of the anti-IL-13 antibody.

## Author contributions

C.-Y.Y. conducted experimental design, performed animal studies, flow cytometric analysis, wrote and edited the manuscript. A.R., A.M., W.X., and M.H. performed research and analyzed data. A.M. provided critical reagents. A.S. assisted with the biostatistical analysis. C.W. performed electron microscopy. N.S. assisted with the design of the study, analyzed data and edited the manuscript. W.R.D. developed the overall concept, designed experiments, supervised the study, and wrote the manuscript.

## Competing interests

W.R.D. receives research support from Sun Pharmaceuticals. The remaining authors declare no conflict of interest.
