## [Peer Review File · Nature Communications]

INTERLEUKIN-9 PRODUCTION BY TYPE 2 INNATE LYMPHOID CELLS INDUCES PANETH CELL METAPLASIA AND SMALL INTESTINAL REMODELINGREVIEWER COMMENTS

Reviewer #1 (Remarks to the Author):

Remarks to the author

Major Comments

(01) CML is a unique driving force in this intestinal remodeling but unfortunately the authors did not emphasize this in their model. Regarding this, the authors should characterize lymphoid/myeloid cell populations in CML mice in more details, a) the ileum vs colon, b) in particular the expansion of various myeloid cells in lamina propria (not just mLN) that could be the initiating cause of epithelial remodeling. The positioning of mast cell in Fig.8 looks like a consequence of ILC2-IL-9 (also indicated in Line 176). As mast cells are also a myeloid cell type so the authors should consider whether mast cells are a driving force once CML is triggered. Please mark CML in Fig.8 to be a major player in this study.

The authors did not provide a clue about how CML-induced GI inflammation (mainly a global myeloid cell expansion) promotes the epithelial increase of IL-33 (Fig.7C-E, H-I) and IL-25 (Fig.4J-K), but not TSLP (Fig.S3B, actually decreased in CML mice), leading to epithelial remodeling. Is mast cell hyperplasia the cause or the consequence of CML? It appears to be the consequence (Line 176 and Fig.8 model). Then what is the cause of increased epithelial IL-33 in the condition of CML, which is not illustrated in the Fig.8 model. I view this as an important and novel point in this study as all other findings (IL-33-ILC2-IL-9, Mast-IL-33 cleavage, IL-9-producing ILC2, IL-9-Paneth cell hyperplasia) are all previously reported.

(02) IL-33, IL-25, and TSLP are all known potent upstream activators of ILC2 and can be all produced by epithelial cells (Line 271-273). In particular, the role of IL-25 in ILC2 activation in CML mice should be further evaluated by IL-25/Tuft cell blockade, and updated in Fig.8.

(03) In Fig.2 and Fig.S1, the authors should show the effect of IL-9 blockade on CML phenotype. Specifically, will IL-9 blockade reduce, a) any myeloid cell expansion in CML mice (i.e. various measurements in spleen, colon, and ileum LP compartments), especially # and % of mast cells and neutrophils by flow cytometry, b) IL-9 levels by ELISA/QPCR in various tissue samples. These analyses will strengthen the causal link between "IL-9-mediated PCM" and "CML" via mast cells (Line 135-138, Line 176-177).

(04) It appears that myeloid cell (especially CD11b+ Gr-1+ neutrophils) expansion is a major aberrant phenotype in CML mice (Line 108-112) (Fig.S1F-G). How about the expansion of neutrophils in the lamina propria of colon vs ileum in CML mice? It has been shown that IL-33 can also be processed into mature bioactive forms by neutrophil elastase and cathepsin (Ref-43), such that neutrophil hyperplasia could also be the reasonable driver (if they were greatly expanded in LP) for the increase of IL-33 bioactive forms. In addition to using many QPCR to assay mast cells (Fcer1-a, mMCP), the authors should show and compare the % and absolute numbers of neutrophils vs mast cells in the lamina propria of colon vs ileum in CML mice, which can be easily done by flow cytometry. Of note, anti-ST2 did not reduce high mMCP levels in CML mice (Fig.7P), suggesting that a) a possible role for neutrophils in regulation of the IL-33-ILC2 axis, and b) the IL-33-ILC2-IL-9-mast cell axis is not operative (can further validate this by a mast cell blockade if possible). Also, can anti-ST2 reduce the % and absolute numbers of neutrophils vs mast cells?

(05) IL-25 mRNA was not increased in the colon of CML mice (Fig.S3A), but was increased in the ileum of CML mice (Fig.4J-K, 5H, 7P). With this, the authors can not exclude a role for IL-25 in ILC2 activation in their model shown in Fig.8, especially the Tuft-IL-25-ILC2 circuit for SI remodeling has been previously shown (Nature 529:221, 2016, Nature 529:226, 2016, Cell 174:271, 2018). Based on this possibility, an increase of Dclk+ Tuft cells was shown in Fig.4J,4K,4M,5H,7P, indicating that Tuft-IL-25 could be a potential upstream driver for ILC2 activation in CML mice. Accordingly, the authors should block IL-25 (compare to IL-33 blockade) in CML mice to evaluate the contribution of IL-33-ILC2 vs IL-25-ILC2 pathways to IL-9 production in CML mice. Of note, anti-ST2 did not reduce high IL-25 levels in CML mice (Fig.7P), this indirectly supports that the IL-25-ILC2 axis could be involved. Additionally, even with IL-33/ST2 blockade, there were still significant Type-2 response (IL-9, IL-4, IL-5, IL-13 in Fig.7N) in CML mice,

suggesting a role for IL-25 in epithelial remodeling.

(06) The authors showed a good correlation of mMCP (mast cell-associated chymase) with levels of bioactive IL-33. However, current data can not support that it is acutally mast cells to cleave full-length IL-33 in their model (especially with the data showing that anti-ST2 can not reduce high mMCP levels in CML mice in Fig.7P, suggesting the ILC2-IL-9-Mast cell axis in Fig.8 is not operative), which requires additional data such as mast cell blockade to make a causal link. Of note, neutrophils can also process full-length IL-33 to bioactive IL-33 (Ref-43).

(07) Features of spontaneous airway inflammation in mice with transgenic expression of IL-9 are abrogated in the progeny of those mice with IL-13 deficiency, indicating that the function of IL-9 might be indirect (via IL-13 and/or IL-5). Of note, IL-5 and IL-13 promote classical features of the Type 2 immune response, such as eosinophil recruitment, mucus production and mast-cell accumulation (Ref-30, Ref-20, Int Immunol 19:1, 2006, J Immunol 178:3244, 2007). How the authors validate this in their model?

Minor Comments

(01) The authors should indicate whether each symbol in each bar graph represents a sample from a single mouse throughout all figures.

(02) Line 809, "the number of Paneth cells per 10 high power fields" is based on what staining? H&E or Phloxine/tartrazine staining? Please indicate this in the legend.

(03) As the numeration of Paneth cells is a key readout throughout the study, it is advised to validate this by using additional approaches (WB-based or Flow-based?) to indicate PCM. Numeration by images sometimes is too selective and not convincing enough.

(04) As IL-9 is a key mediator in this sudy, the IL-9 protein measurement by ELISA (and after IL-9 blockade) should be provided here in Fig.2A.

(05) The effects of anti-IL-9 to block Paneth cells in the ileum of CML mice were not consistent (right panels in Fig.2E/2F vs Fig.2C). Similarly, the effects of anti-IL-13 to block Paneth cells in the ileum/colon of CML mice were not consistent (Fig.5B vs Fig.5C). This raises the concern (as described in (3) above) that the numeration of Paneth cells by images appears not very correlated to QPCR of Paneth cell marker Crypt 1 and sPLA2.

(06) As small intestine hyperplasia appears well-studied in recent years and pretty much cell types (both epithelial cells and ILC), or cytokine (IL-33, IL-25, TSLP, IL-4/5/15, IL-9, IL-2) involved in the remodeling are characterized and established. As such, this reviewer do not understand why the authors will go for metabolic profiling but not RNA seq or cytokine profiling to identify cells and cytokines involved in CML in the beginning.

(07) WB data quality need to be improved. a) WB of Fig.7H appears not on the same blots. To calculate the statistical significance as shown in Fig.7I, samples of Tet ON and CML need to be presented or loaded on the same blot. b) Intensity of IL-33 (35kDa) from ileum CML appears very different in Fig.7J (upper band?) vs Fig.S3C.

(08) Data in Fig.6H is mLN and colon, but not ileum and colon, as described in Line-259.

(09) Line 362-364, whether Ref-30 is related to the statement here needs to be validated.

(10) IL-25, IL-33, TSLP or a combination of these cytokines did not induce substantial IL-9 production in ILCs but culturing ILCs in the presence of IL-2 led to IL-9 protein expression (Ref-35). While it may not be the focus of this study, the role of IL-2 in helping IL-9 production by ILC2 is not discussed here. The reason why I mention IL-2 here is that IL-2 could be produced by dendritic cells, mast cells, or maybe other myeloid cells expanded in CML mice. The IL-33+CML-derived IL-2=ILC2-IL-9 axis could indeed serve as a good mechanism driven by a CML-derived factor. In other words, IL-2 could be the missing factor linking CML to ILC2 activation.

Reviewer #2 (Remarks to the Author):

In this study, Yuan and colleagues provide new data on the development of Paneth cell metaplasia (PCM) in the context of an extra intestinal disease. i.e. chronic myelogenous leukemia (CML) using a mouse model. They found that PCM can be initiated in the absence of a pre-existing gastrointestinal disorder. In addition, they were able to identify the molecular mechanism involved in PCM by gene expression and immunofluorescence techniques. They propose that the IL-33/ILC2/IL-9 axis is responsible for the intestinal remodelling phenotype, suggesting a relevant role for development of PCM in gastrointestinal disorders such as colorectal cancer and IBD. Overall, the manuscript is well written and the results are of potential interest. However, I feel that further data are required to strengthen their conclusions.

Figure 1:

- I would recommend the authors to perform IF staining for Lysozyme together with E-cad or Villin in order to validate the PCM phenotype in CML mice.
- Furthermore, it would be interesting to assess, if cell proliferation is also increased.
- In Fig 1 J Day 0 (as control) is missing.
- In Fig 1 K please provide tissue images showing the resolution of PCM

Figure 2:

- IL-9 measurements are provided only by RNA quantification. The authors need to show IL-9 protein expression/ production.
- The number of Paneth cell in the isotype control seems to be lower compared to the phenotype described in Figure 1. The authors should comment how reproducible/ variable the PCM phenotype in the CML model is.
- The IL-9 antibody experiment is a key piece of data for the study. Thus, it would be good to further strengthen those data by additional readout. Furthermore, it would be good to confirm the role of IL-9 in a second PCM model.

Figure 4:

- The authors observed an increased small intestinal length; Thus, it would be interesting to measure cell proliferation. Furthermore, is Tuft cell and Goblet cell hyperplasia also observed in the colon?
- I would be important to assess the functional relevance of Tuft cell and Goblet cell hyperplasia? Do the mice have a compromised gut barrier? Is the mucus layer affected? Do the mice develop spontaneous inflammation?

Figure 5: the authors studied the influence of IL-13 on PCM; it would be interesting to also assess the role of IL-5.

Figure 6: authors should provide a better characterization of the source of IL-9 in the small intestine, colon and mLN of CLM mice, e.g. by using flow cytometry. This is an important conclusion of the study, but the provided evidence appears to be weak.

The authors conclude: "Collectively, these data identified type 2 innate lymphoid cells (ILC2s) as the primary source of IL-9 within the ileum and colon of CML mice". ILC2 increase is shown only in the colon and not in small intestine and it is rather subtle. Thus, further functional data sustaining this conclusion would be important.

Figure 7:

- IL-33 receptor (ST2) has been described to be lowly expressed in intestinal ILC2 (Dutton et al., 2018). Thus, it would be interesting to test, how ST2 expression in ILC2 is modulated in the CML model.
- The authors used a blocking ST2 antibody. However, this causes a global and not an ILC2 specific ST2 blockade. The authors should however test, if IL-33 acts directly on ILC2.

Reviewer #3 (Remarks to the Author):

Authors described the development of Paneth cell metaplasia(PCM) and intestinal remodelling in a murine BCR::ABL1 inducible model of Chronic Myeloid Leukaemia (CML). Mechanistically, CML induced a pro-inflammatory status in the gastrointestinal tract which resulted in the production of IL-33 by epithelial cells. This led to the activation of a type 2 innate lymphoid cells (ILC2s) which in turn resulted in the production of IL-9 which directly remodelled the small intestine characterized by lengthening with goblet and tuft cell hyperplasia. In conclusion, authors showed that intestinal inflammation can trigger an ILC2/IL-9 circuit that induces PCM and regulates the development of specialized epithelial cell populations resident in the gastrointestinal tract in a mouse model of CML.

Despite the mechanistic pathway by which PCM develop is very interesting, several questions have emerged.

- 1- Is it not clear the connection between PCM and CML. Which is the rational for choosing CML as a model of PCM? This must be addressed more in detail, because to my knowledge PCM has not been described on this disease. I understand that this is a model, but it needs to be explained.
- 2- Authors should state clearly that off-tet is CML and to use the same nomenclature in all text.
- 3- Discussion line 306: Authors stated "to validate a causative relationship between PCM and CML, we employed a tet mouse modeland BCR::ABL1" The causative relationship is still not clear, ad example how BCR::ABL1 is inducing PCM. Authors are describing an association. Authors should try to engraft mouse with primary cells from CML patients and verify PCM.
- 4- It is not clear the translation value of this value for CML patients.

- 5- This mouse model is a very specific model driven by BCR::ABL1, and this pathway should be validated in another model, for example in intestinal organoids in a model without BCR::ABL1.

- 6- Because of the model used, it is still not totally clear the translation value in the context of CML, especially following author's conclusions stating "IL-33-regulated ILC2/IL-9 immune circuit... may have pathophysiological relevance for the emergence of PCM in diseases as colorectal cancer and inflammatory bowel disease" at line 416..however this has not been proven on these settings. Additional evidence is needed.

The individual comments of each of the reviewers have been addressed in a point-by-point fashion as detailed below.

The comments of Reviewer 1 were addressed as follows:

1. *CML is a unique driving force in this intestinal remodeling but unfortunately the authors did not emphasize this in their model. The authors should characterize lymphoid/myeloid cell populations in CML mice in more details, a) the ileum vs colon, b) in particular the expansion of various myeloid cells in lamina propria (not just mLN) that could be the initiating cause of epithelial remodeling. The positioning of mast cell in Fig.8 looks like a consequence of ILC2-IL-9 as mast cells are also a myeloid cell type so the authors should consider whether mast cells are a driving force once CML is triggered. Please mark CML in Fig.8 to be a major player in this study. The authors did not provide a clue about how CML-induced GI inflammation (mainly a global myeloid cell expansion) promotes the epithelial increase of IL-33 and IL-25, but not TSLP (actually decreased in CML mice), leading to epithelial remodeling. Is mast cell hyperplasia the cause or the consequence of CML? It appears to be the consequence (Line 176 and Fig.8 model). Then what is the cause of increased epithelial IL-33 in the condition of CML, which is not illustrated in the Fig.8 model. I view this as an important and novel point in this study as all other findings (IL-33-ILC2-IL-9, Mast-IL-33 cleavage, IL-9-producing ILC2, IL-9-Paneth cell hyperplasia) are all previously reported.*

To address this comment by the reviewer, we have analyzed myeloid cell expansion in the colon of CML mice when compared to non-leukemic controls animals. The results of this analysis demonstrated that macrophages, dendritic cells and neutrophils were all significantly increased in the lamina propria of the colon. Thus, there is general expansion of the myeloid compartment with the most noteworthy increase occurring in neutrophils which is not surprising since this is the most prominent characteristic of CML. These data are now depicted as new **Figure 7B**.

We have also noted the role of CML and associated inflammatory cytokine production as critical factors in the development of intestinal remodeling in the **Figure 9** (formerly Figure 8) schematic as requested by the reviewer.

The reviewer raises an excellent point about further definition of the upstream trigger for IL-33 production. Cellular stress and damage are a well-recognized stimulus for IL-33 release from cells, and we have postulated that CML induces an inflammatory state that results in cellular damage. To provide confirmation for this premise, we assessed cleaved caspase 3 expression since this is a marker for apoptotic cell death and therefore a direct indicator of cellular damage. Employing both immunofluorescence staining of ileal and colonic tissue along with western blot analysis of lysates from these same tissues, we now demonstrate that cleaved caspase 3 expression is significantly increased in the intestinal epithelial cell layer of CML mice. Thus, these data provide further support for the premise that an inflammatory cytokine milieu that arises in the GI tract of CML mice leads to cellular damage and death, thereby promoting the production of IL-33 by damaged intestinal epithelial cells. These results are now shown as new data in **Figures 7E-7G**.

2. *IL-33, IL-25, and TSLP are all known potent upstream activators of ILC2 and can be produced by epithelial cells). In particular, the role of IL-25 in ILC2 activation in CML mice should be further evaluated by IL-25/Tuft cell blockade and updated in Fig.8.*

To address this question, we had to obtain a blocking anti-IL-25 antibody that was suitable for in vivo administration. To our knowledge, there is only one such antibody in the world which was

made in the lab of Dr Andrew Mackenzie from the United Kingdom. After a considerable amount of time, we were able to obtain the hybridoma through a collaborative Material Transfer Agreement. We subsequently cultured the hybridoma, collected the culture supernatant, and purified anti-IL-25 antibody using a protein G column. To determine whether IL-25 has an upstream role in the activation of ILC2 and the development of PCM, we then conducted experiments in which CML mice were treated with an anti-IL-25 or an isotype control antibody three times per week. Results of these experiments revealed that blockade of IL-25 signaling had no effect on gene expression of IL-13 or IL-9 in the ileum nor was there any difference in the expression of crypt1 or sPLA in the ileum or colon. In addition, antibody blockade of IL-25 did not alter gene expression of mMCP 1,2,4 in mast cells or DCLK-1 in tuft cells, indicating that IL-25 had no apparent role in the upstream regulation of IL-13, or the expression of genes associated with specialized epithelial cell populations in CML mice. These results are now depicted as new data in **Figures 5J-5N**.

3. *In Fig.2 and Fig.S1, the authors should show the effect of IL-9 blockade on CML phenotype. Specifically, will IL-9 blockade reduce, a) any myeloid cell expansion in CML mice (i.e., various measurements in spleen, colon, and ileum LP compartments), especially # and % of mast cells and neutrophils by flow cytometry, b) IL-9 levels by ELISA/QPCR in various tissue samples. These analyses will strengthen the causal link between “IL-9-mediated PCM” and “CML” via mast cells (Line 135-138, Line 176-177).*

To address the effect of IL-9 signaling blockade on the CML phenotype, lethally irradiated FVB mice were transplanted with BM cells from SCLtTa / bcr-abl animals. Cohorts of mice were maintained on or taken off Tet and then treated with an isotype control or anti-IL-9 antibody three times per week. Results of these experiments which are now depicted in new **Supplemental Figure 2** demonstrated that animals that were treated with anti-IL-9 antibody had increased weight loss (**Supplemental Figure 2A**) and a significant time-dependent increase in white blood cell and granulocyte counts in the peripheral blood when compared to mice that received an isotype control antibody (**Supplemental Figures 2B and 2C**). While there was no difference in spleen weight between groups (**Supplemental Figure 2D**), the absolute number of myeloid CD11b⁺ Gr-1⁺ cells were also significantly augmented in mice that were treated with anti-IL-9 antibody (**Supplemental Figure 2E**). Thus, these results demonstrated that blockade of IL-9 signaling resulted in an increased number of granulocytes in the spleen and blood, indicating that IL-9 appears to have some regulatory effects on CML progenitor populations and the maturation pathway by which these cells become terminal granulocytes. How this occurs is an interesting hematological question but is beyond the scope of the present study.

4. *It appears that myeloid cell (especially CD11b⁺ Gr-1⁺ neutrophils) expansion is a major aberrant phenotype in CML mice (Fig.S1F-G). How about the expansion of neutrophils in the lamina propria of colon vs ileum in CML mice? It has been shown that IL-33 can also be processed into mature bioactive forms by neutrophil elastase and cathepsin (Ref-43), such that neutrophil hyperplasia could also be the reasonable driver for the increase of IL-33 bioactive forms. In addition to using many QPCR to assay mast cells (Fcer1-a, mMCP), the authors should show and compare the % and absolute numbers of neutrophils vs mast cells in the lamina propria of colon vs ileum in CML mice, which can be easily done by flow cytometry. Of note, anti-ST2 did not reduce high mMCP levels in CML mice (Fig.7P), suggesting that a) a possible role for neutrophils in regulation of the IL-33-ILC2 axis, and b) the IL-33-ILC2-IL-9-mast cell axis is not operative (can further validate this by a mast cell blockade if possible).*

As noted in our response to Comment #1, we have now characterized the myeloid cell expansion in the GI tract of CML mice (**Figure 7B**). Not unexpectedly, CML had a significant increase in

the frequency of neutrophils in the colons when compared to non-leukemic controls. There was also an increase in the percentage of macrophages and dendritic cells, but this was much more modest. In **Figures 4C and 4D**, we demonstrated that mast cell hyperplasia in the GI tract of CML mice was attributable to mucosal as opposed to connective tissue mast cells (that are located in the lamina propria) based on the type of mast cell protease that was produced. Therefore, we could not quantify mucosal mast cell numbers in the lamina propria (as we did for neutrophils) because they reside exclusively in the epithelial cell layer of CML mice (**Figures 4E and 4F**).

The reviewer raised an excellent point regarding the potential role of neutrophil elastase and cathepsin G as potential mediators of IL-33 cleavage into the more mature short form since these enzymes are produced by neutrophils. To address this question, we performed western blot analysis and observed that there was an increase in the expression of cathepsin G and neutrophil elastase in the ileum and colon of CML versus non-leukemic control mice (**new Figures 8A-8C**). Notably, however, administration of anti-IL-9 antibody resulted in a reduction in neutrophil elastase only in the colon and there were only modest effects of antibody administration on cathepsin G levels in this same tissue site, with no effect on either protein in the ileum (**new Figures 8F, 8I and 8J**). In contrast, blockade of IL-9 signaling significantly reduced expression of mast cell chymase in both ileum and colon (**Figure 8K**). This is noteworthy since anti-IL-9 antibody administration reduced levels of the short more potent form of IL-33 in both tissue sites as well, indicative of a more direct correlation between chymase and the 15kDA IL-33 product. Therefore, while these data suggest that cathepsin G and neutrophil elastase could also have a role in mediating cleavage of IL-33 into more mature forms, we think it most likely that they play a more minor role, and that mast cells remain the most likely population responsible for IL-33 cleavage. However, to account for these new data, we have modified the **Figure 9** schematic to include a role, albeit presumed to be more minor, for neutrophils in the degradation of IL-33 to smaller fragments.

5. *IL-25 mRNA was not increased in the colon of CML mice (Fig.S3A) but was increased in the ileum of CML mice (Fig.4J-K, 5H, 7P). With this, the authors cannot exclude a role for IL-25 in ILC2 activation in their model shown in Fig.8, especially the Tuft-IL-25-ILC2 circuit for SI remodeling has been previously shown (Nature 529:221, 2016, Nature 529:226, 2016, Cell 174:271, 2018). Based on this possibility, an increase of Dclk+ Tuft cells was shown in Fig.4J,4K,4M,5H,7P, indicating that Tuft-IL-25 could be a potential upstream driver for ILC2 activation in CML mice. Accordingly, the authors should block IL-25 (compare to IL-33 blockade) in CML mice to evaluate the contribution of IL-33-ILC2 vs IL-25-ILC2 pathways to IL-9 production in CML mice.*

This reviewer comment regarding a potential role for IL-25 as an upstream regulator of ILC2s and the development of PCM is similar to that raised in Point #2. In response, we conducted in vivo IL-25 antibody blockade studies as requested and now present these results in **Figures 5J-5N**. More specific details are noted in our response to Point #2.

6. *The authors showed a good correlation of mMCP (mast cell-associated chymase) with levels of bioactive IL-33. However, current data cannot support that it is actually mast cells to cleave full-length IL-33 in their model (especially with the data showing that anti-ST2 cannot reduce high mMCP levels in CML mice in Fig.7P, suggesting the ILC2-IL-9-Mast cell axis in Fig.8 is not operative), which requires additional data such as mast cell blockade to make a causal link. Of note, neutrophils can also process full-length IL-33 to bioactive IL-33 (Ref-43).*

We agree with the reviewer that neutrophil-derived neutrophil elastase or cathepsin G could also play a role in the cleavage of IL-33 to shorter more mature forms. In response to an earlier query by the reviewer as to this point, we addressed this in our response to Reviewer Comment #4 with new data provided in the manuscript (**Figures 8A-8C**). With respect to the question of mast cell

blockade, the only potential mast cell blocking strategy that we are aware of is the use of cromolyn, but this agent inhibits degranulation of histamine and leukotrienes, whereas any effect on mast cell proteases has not been well studied. Thus, we do not know of any targeted pharmacological or antibody-based approaches that specifically block the production of mast cell proteases. Nonetheless, we would note that IL-9 signaling blockade had more profound effects on mast cell chymase (**Figure 8K**) than either of the neutrophil-derived proteases (**Figures 8I and 8J**). For that reason, we believe that mast cells remain the primary IL-9 responsive population in this model, although we cannot exclude a more reduced role for neutrophils and have therefore included them in our updated **Figure 9** schematic.

7. *Features of spontaneous airway inflammation in mice with transgenic expression of IL-9 are abrogated in the progeny of those mice with IL-13 deficiency, indicating that the function of IL-9 might be indirect (via IL-13 and/or IL-5). Of note, IL-5 and IL-13 promote classical features of the Type 2 immune response, such as eosinophil recruitment, mucus production and mast-cell accumulation (Ref-30, Ref-20, Int Immunol 19:1, 2006, J Immunol 178:3244, 2007). How the authors validate this in their model?*

We appreciate the author's comment regarding whether some of the actions of IL-9 could be mediated through either IL-13 or IL-5. With respect to IL-13, we did examine the putative role of IL-13 and found that IL-13 regulated Paneth cell-specific gene expression along with tuft cell and goblet cell hyperplasia but had no effect on the development of Paneth cell metaplasia or hyperplasia during CML (**Figure 5**). Thus, IL-13 regulated tuft and goblet cell development in CML mice but had no effect on the emergence of PCM or mast cell hyperplasia, indicating that IL-13 only partially accounted for the effects observed after IL-9 blockade. With respect to IL-5, we have conducted additional experimentation in which lethally irradiated FVB mice were transplanted with BM from SCLT^a/bcr-abl mice. Animals were either maintained on or taken off Tet and then treated two times per week with an isotype control or anti-IL-5 antibody, similar to the experimental approach employed to assess the role of IL-9 and IL-13 antibody blockade. Results of these studies demonstrated that blockade of IL-5 signaling had no effect on expression of the Paneth cell markers, crypt 1 and sPLA2, IL-9, IL-13, mMCP1,2,4, IL-25, or DCLK-1. Thus, these data indicate that, in contrast to IL-13, IL-5 had no role in the development of PCM or regulation of specialized epithelial cell populations in this model. These results are now depicted in **Supplemental Figure 3**

Minor Points:

1. *The authors should indicate whether each symbol in each bar graph represents a sample from a single mouse throughout all figures.*

We have stated that each symbol in each of the scatterplots are from individual mice.

2. *Line 809, "the number of Paneth cells per 10 high power fields" is based on what staining? H&E or Phloxine/tartrazine staining? Please indicate this in the legend.*

We now indicate in the legend that quantitation of Paneth cells was performed based on H&E staining.

3. *As the numeration of Paneth cells is a key readout throughout the study, it is advised to validate this by using additional approaches (WB-based or Flow-based?) to indicate PCM. Numeration by images sometimes is too selective and not convincing enough.*

We appreciate the reviewer's point with respect to Paneth cell quantitation. We considered other approaches but elected to employ H&E staining to assess Paneth cell quantitation because we thought that other approaches, such as western blot imaging and flow cytometry, would be less precise. Specifically, we did not think quantitation of defensins that reside within Paneth cell by western blot analysis would not be an accurate indicator of total Paneth cell numbers which is the primary endpoint. Secondly, to our knowledge, there are no known specific cell surface markers expressed solely on Paneth cells that could be employed to enumerate these cells by flow cytometry. We are cognizant of the need for accurate Paneth cell enumeration and for that reason would point out that we consistently counted at least 10 high power fields per tissue section and did replicate experiments for each experimental condition to ensure that the results were accurate. We also performed Paneth cell-specific lysozyme immunofluorescence staining in **Figure 1E** to further validate the increased number of Paneth cells.

4. *As IL-9 is a key mediator in this study, the IL-9 protein measurement by ELISA (and after IL-9 blockade) should be provided here in Fig.2A.*

We attempted to measure IL-9 in the serum and tissue explants by bioplex but observed that there was no detectable circulating IL-9. This is likely due to very low circulating levels in the blood that are below the level of detection of the assay. However, as an alternative approach, to provide a more quantitative assessment of IL-9 in a relevant tissue site, we took advantage of the fact that our immunofluorescence staining had demonstrated IL-9 protein expression in both the colon and ileum of CML mice. Therefore, we re-examined these tissues and performed a very detailed quantitative enumeration (as detailed in the Methods section of the manuscript) of IL-9 expressing cells along with associated lineage-defining markers. The results of this analysis which is now depicted as new **Figure 6H** demonstrated that an average of 92% and 95% of all IL-9⁺ cells in the ileum and colon, respectively, co-expressed GATA3 but lacked expression of CD3, defining them as ILC2s. In contrast, only about 5% of cells had a TH2 phenotype (i.e., CD3⁺ GATA3⁺) and virtually none of these cells were CD3⁻ GATA3⁻. Thus, these data provide further evidence of protein expression of IL-9 by type 2 ILCs in the GI tract (i.e., ileum and colon).

5. *The effects of anti-IL-9 to block Paneth cells in the ileum of CML mice were not consistent (right panels in Fig.2E/2F vs Fig.2C). Similarly, the effects of anti-IL-13 to block Paneth cells in the ileum/colon of CML mice were not consistent (Fig.5B vs Fig.5C). This raises the concern (as described in (3) above) that the numeration of Paneth cells by images appears not very correlated to QPCR of Paneth cell marker Crypt 1 and sPLA2.*

The reviewer notes that there was some discordance observed with respect to the effect of anti-IL-9 or anti-IL-13 antibody blockade on histological evidence of PCM versus gene expression of specific Paneth cell antimicrobial peptides. In response to this concern, we would note that the number of granules which contain anti-microbial proteins can be variable in Paneth cells and are not always concordant with the absolute number of these cells. Relevant to this point are two studies by Stockinger and colleagues (*Interleukin 13 mediated Paneth cell degranulation and antimicrobial peptide release, J Innate Immunity, 2014*) and Murano and colleagues (*Transcription factor TFEB-cell autonomously modulates susceptibility to intestinal epithelial cell injury, Scientific Reports, 2017*) who both showed that, under certain experimental conditions, Paneth cell numbers were not concordant with the number of granules (i.e., cells may be depleted of granules

in select instances). In the specific examples mentioned, we would note that in Figure 2, this was only observed when comparing the absolute number of Paneth cell in the ileum with gene expression of crypt 1 and sPLA2 in this tissue site, whereas results in the colon were completely concordant. So, in this instance, we believe the overall conclusion that IL-9 blockade prevented PCM was still valid. In the second instance, the discordance between Paneth cell numbers and gene expression of specific anti-microbial granules after IL-13 blockade served to highlight that IL-13 did not phenocopy the results observed with IL-9 and thereby was downstream of this cytokine as it only had consistent effects on tuft and goblet cells which was what we concluded.

6. *As small intestine hyperplasia appears well-studied in recent years and pretty much cell types (both epithelial cells and ILC), or cytokine (IL-33, IL-25, TSLP, IL-4/5/15, IL-9, IL-2) involved in the remodeling are characterized and established. As such, this reviewer does not understand why the authors will go for metabolic profiling but not RNA seq or cytokine profiling to identify cells and cytokines involved in CML in the beginning.*

Paneth cells produce defensins and other molecules that are important in protecting the host from bacterial pathogens and in regulating the composition of intestinal flora. We therefore considered that the development of PCM in the colon and Paneth cell hyperplasia in the ileum, which led to the increased production of anti-microbial peptides, might affect the production of a wide array of metabolites by host cells given the immune modulatory properties of these peptides. Consequently, we reasoned the metabolic profiling would be more likely to uncover such differences as opposed to RNA sequencing studies since this approach would identify the specific proteins or lipid moieties that were differentially expressed between groups, and not just transcriptional differences which are not always concordant with protein analysis.

7. *WB data quality need to be improved. a) WB of Fig.7H appears not on the same blots. To calculate the statistical significance as shown in Fig.7I, samples of Tet ON and CML need to be presented or loaded on the same blot. b) Intensity of IL-33 (35kDa) from ileum CML appears very different in Fig.7J (upper band?) vs Fig.S3C.*

The western blot data shown in **Figure 7H (which is now Figure 7K)** was actually from the same blot; the intensity difference was due to the fact that a smaller amount of protein was loaded in the far three lanes because the intensity of the IL-33 bands in the colon was so bright in CML mice. Since actin served as the internal control, we were still able to normalize the data appropriately. The complete blot is now depicted in **new Supplemental Figure 5K**. With respect to **Figure 7J (which is now Figure 8F)** versus **Supplemental Figure 3C (which is now Supplemental Figure 5C)**, these were samples from animals in different experiments harvested at different time points which is why the intensities between these two blots differed.

8. *Data in Fig.6H is mLN and colon, but not ileum and colon, as described in Line-259.*

We have corrected the discrepancy as pointed out after careful examination by the reviewer.

9. *Line 362-364, whether Ref-30 is related to the statement here needs to be validated.*

We thank the reviewer for pointing out that the reference citation did not align with the sentence. We have removed the citation in the text.

10. *IL-25, IL-33, TSLP or a combination of these cytokines did not induce substantial IL-9 production in ILCs but culturing ILCs in the presence of IL-2 led to IL-9 protein expression (Ref-35). While it*

may not be the focus of this study, the role of IL-2 in helping IL-9 production by ILC2 is not discussed here. The reason why I mention IL-2 here is that IL-2 could be produced by dendritic cells, mast cells, or maybe other myeloid cells expanded in CML mice. The IL-33+CML-derived IL-2-ILC2-IL-9 axis could indeed serve as a good mechanism driven by a CML-derived factor. In other words, IL-2 could be the missing factor linking CML to ILC2 activation.

We appreciate this point made by the reviewer and agree that other cytokines such as IL-2 might serve to potentiate activation of ILC2s as described in other publications. We have therefore modified the Discussion to state that it possible that other cytokines such as IL-2 could act in concert with IL-33 to promote production of IL-9 leading to the downstream development of PCM and intestinal remodeling.

The comments of Reviewer 2 were addressed as follows:

1. *I would recommend the authors to perform IF staining for Lysozyme together with E-cad or Villin in order to validate the PCM phenotype in CML mice. Furthermore, it would be interesting to assess, if cell proliferation is also increased. In Fig 1 J Day 0 (as control) is missing. In Fig 1 K please provide tissue images showing the resolution of PCM.*

We appreciate the reviewer's detailed comments for additional clarification as to data presented in Figure 1. As requested, we have performed immune fluorescence staining of the colon for lysozyme which further confirms Paneth cell metaplasia (PCM) in CML mice. These data are now shown in modified **Figure 1E**. To assess cell proliferation in the intestinal tract, we administered BrdU to CML mice 28 days post transplantation at a time when PCM is well developed. Results of these studies demonstrated that there was significantly increased BrdU incorporation into intestinal epithelial cells in CML when compared to non-CML animals. These results now appear as modified **Figures 4H and 4I**. We did not include a day 0 WBC measurement since these are essentially normal mice (prior to irradiation and transplantation) and have white blood cell counts within the normal range that we highlighted by the horizontal lines in the Figure. We have also provided tissue images showing resolution of PCM in the Tet ON group and persisting metaplasia in the CML (Tet OFF) group. This new data is depicted in modified **Figure 1L**.

2. *IL-9 measurements are provided only by RNA quantification. The authors need to show IL-9 protein expression/ production. The number of Paneth cell in the isotype control seems to be lower compared to the phenotype described in Figure 1. The authors should comment how reproducible/ variable the PCM phenotype in the CML model is. The IL-9 antibody experiment is a key piece of data for the study. Thus, it would be good to further strengthen those data by additional readout. Furthermore, it would be good to confirm the role of IL-9 in a second PCM model.*

To assess IL-9 protein expression, we attempted western blot analysis of ileum and colon tissue lysates from CML mice but observed that this approach was not sufficiently sensitive to detect IL-9 production. Therefore, we pursued an alternative approach which took advantage of the fact that our immunofluorescence staining demonstrating IL-9 in the colon and ileum of CML mice was in fact evidence of IL-9 protein expression. Therefore, we re-examined these tissues and performed a very detailed quantitative enumeration (as detailed in the Methods section of the manuscript) of IL-9 expressing cells along with associated lineage-defining markers. The results of this analysis which is now depicted as **Figure 6H** in modified Figure 6 demonstrated that an average of 92% and 95% of all IL-9⁺ cells in the ileum and colon, respectively, co-expressed GATA3 but lacked expression of CD3, defining them as ILC2s. In contrast, only about 5% of cells had a TH2

phenotype (i.e., CD3⁺ GATA3⁺) and virtually none of these cells were CD3⁻ GATA3⁻. Thus, these data provide evidence of protein expression of IL-9 by type 2 ILCs.

The reason that the Paneth cell counts in the colon in **Figure 2C** are lower than those enumerated in **Figure 1F** (previously Figure 1E) is because mice in Figure 2 were analyzed at 21 days post transplantation, whereas animals in Figure 1 were examined 35 days after transplantation. The reason for this is that mice in Figure 2 were treated with an anti-IL-9 antibody which resulted in augmented weight loss and a significant increase in granulocytosis in the peripheral blood and spleen (see **new Supplemental Figure 2** in response to Reviewer 1 query). Therefore, these mice were more ill, and we elected to examine animals in this experimental cohort at an earlier time point. Thus, PCM as determined by H&E staining was not as evident on day 21 when compared to day 35, although the differences between isotype and anti-IL-9 treated groups were still statistically significant. In addition, we would point out that the gene expression values of antimicrobial peptides in **Figures 2E and 2F** were consistent with what was observed in **Figures 1H and 1I** when assessed at the same time point (i.e., 21 days).

With respect to the confirmation of IL-9 driven PCM in another model, we extensively reviewed the literature to determine whether there were any other disease specific murine models in which PCM was reported to occur. This review process did not reveal any other disease-oriented models which describe this phenomenon, and therefore would allow us to validate this mechanistic pathway in another setting. We therefore know of no way to address this query. We believe, however, that the absence of prior mouse models only serves to highlight the novelty of these findings. In addition, our report is the first study to delineate a mechanistic pathway by which PCM develops in any disease setting, and therefore provides foundational work if a yet to be described model uncovers the same observation of PCM in the future.

- 3. The authors observed an increased small intestinal length; Thus, it would be interesting to measure cell proliferation. Furthermore, is Tuft cell and Goblet cell hyperplasia also observed in the colon? It would be important to assess the functional relevance of Tuft cell and Goblet cell hyperplasia. Do the mice have a compromised gut barrier? Is the mucus layer affected? Do the mice develop spontaneous inflammation?*

We appreciate the reviewer's comments for clarification as to data presented with respect to intestinal remodeling and have conducted additional experimentation to address these concerns. As noted in the response to Point #1 raised by the reviewer, we have employed BrdU labeling to demonstrate that there is indeed increased proliferation of epithelial cells in the colon and ileum in CML mice (**Figures 4H and 4I**), as a further way to assess the increased small intestinal length that we observed in these animals. Goblet cells are ubiquitous in the colon of normal mice; therefore, it was not possible to accurately quantify whether there was hyperplasia in the colon of CML animals since this cell type is so widely prevalent. Also, tuft cells in the colon have a different transcriptional profile than those in the ileum (*Nadsombati et al, Immunity, see reference 27*), so we focused only on ileal tuft cells since our primary focus was understanding the mechanistic underpinning of intestinal remodeling in this tissue site. To assess the mucus layer in CML versus non leukemic animals, we stained ileal tissue using Carnoy's fixative which is a histological technique that preserves the mucus layer. This analysis demonstrated that there was an increased mucus layer in the ileum of CML mice when compared to non-leukemic control animals. This is now shown as **Figure 4P**. Finally, we performed histopathological analysis on colonic tissue from both groups of animals to examine whether there was any overt pathological damage in CML indicative of a compromised gut barrier or spontaneous inflammation. Results of this analysis revealed no difference in pathological scores between these two cohorts of mice using a semi quantitative scoring system (detailed in the Methods section). This is now shown as **Figure 7A**.

4. *The authors studied the influence of IL-13 on PCM; it would be interesting to also assess the role of IL-5.*

To address the role of IL-5, lethally irradiated FVB mice were transplanted with BM from SCLtTa/bcr-abl mice. Animals were either maintained on or taken off Tet and then treated two times per week with an isotype control or anti-IL-5 antibody, similar to the experimental approach employed to assess the role of IL-9 and IL-13 antibody blockade. Results of these studies demonstrated that that blockade of IL-5 signaling had no effect on expression of the Paneth cell markers, crypt 1 and sPLA2, IL-9, mMCP1, IL-25, DCLK-1 or IL-13. Thus, these data indicate that, in contrast to IL-13, IL-5 has no obvious role in the development of PCM or intestinal remodeling in this model. These results are now depicted in **Supplemental Figure 3**.

5. *The authors should provide a better characterization of the source of IL-9 in the small intestine, colon and mLN of CLM mice, e.g., by using flow cytometry. This is an important conclusion of the study, but the provided evidence appears to be weak. The authors conclude: “Collectively, these data identified type 2 innate lymphoid cells (ILC2s) as the primary source of IL-9 within the ileum and colon of CML mice”. ILC2 increase is shown only in the colon and not in small intestine and it is rather subtle. Thus, further functional data sustaining this conclusion would be important.*

We attempted on several occasions to detect IL-9 expressing cells in the colon and mLN by flow cytometry. Specifically, we employed two directly conjugated anti-IL-9 antibodies and one that was unconjugated and required staining with a secondary fluorochrome conjugated antibody. In each instance, we were not able to detect an IL-9 expressing population which we attributed to the fact that only a very small ILC2 population produces IL-9 and flow cytometry is not sufficiently sensitive to detect this subset. However, as an alternative approach, to provide a more quantitative assessment of IL-9, we took advantage of the fact that our immunofluorescence staining demonstrated IL-9 protein expression in both the colon and ileum of CML mice. Therefore, we re-examined these tissues and performed a very detailed quantitative enumeration (as detailed in the Methods section of the manuscript) of IL-9 expressing cells along with associated lineage-defining markers. As noted in response to Point #2, the results of this analysis which is now depicted as modified **Figure 6H** demonstrated that an average of 92% and 95% of all IL-9⁺ cells in the ileum and colon, respectively, co-expressed GATA3 but lacked expression of CD3, defining them as ILC2s. In contrast, only about 5% of cells had a TH2 phenotype (i.e., CD3⁺ GATA3⁺) and virtually none of these cells were CD3⁻ GATA3⁻. Thus, these data provide evidence of protein expression of IL-9 by type 2 ILCs in both ileum and colon.

6. *IL-33 receptor (ST2) has been described to be lowly expressed in intestinal ILC2 (Dutton et al., 2018). Thus, it would be interesting to test, how ST2 expression in ILC2 is modulated in the CML model. The authors used a blocking ST2 antibody. However, this causes a global and not an ILC2 specific ST2 blockade. The authors should however test if IL-33 acts directly on ILC2.*

We appreciate the reviewer's question as to whether IL-33 acts directly on ILC2s. The only way that we know to directly address this question would be to conditionally delete ST2 from ILC2s and then determine whether this attenuates the development of PCM and intestinal remodeling. This would be an extremely laborious process which would require the creation of a PLZF-Cre ST2 floxed mouse. While this is theoretically feasible, these mice are currently available only on a B6 background whereas the CML model is restricted to FVB background animals. Thus, this process would take over 2 years to complete which, in our view, would be beyond the scope of the current project. With that said, we do believe that the current data in the manuscript provide compelling support for the premise that IL-33 acts directly on ILC2s. First of all, the primary IL-33 responsive

populations in the literature have been shown to be CD4+ T cells (i.e., TH2 and regulatory T cells), mast cells, and ILC2s. We have clearly shown that depletion of CD4+ T cells from CML mice has no effect on gene expression of IL-9 (**Figure 6E and Supplemental Figure 4**) which is the critical cytokine that is absolutely necessary for development of PCM. In addition, we have demonstrated using immunofluorescence staining that mast cells do not express IL-9 protein in the GI tract. Therefore, if IL-33 were to bind to ST2 on mast cells, the effects would appear to be negligible. In contrast, we have demonstrated clearly that ILC2s produce IL-9 protein (**Figures 6F and 6G**) and that blocking ST2 results in a significant decrease in ILC2-derived cytokines (i.e., IL-9, IL-4, IL-5, and IL-13) (**Figures 8L and 8M**). Thus, collectively, we believe that these results provide strong rationale for the conclusion that IL-33 likely directly acts on ILC2s.

The comments of Reviewer 3 were addressed as follows:

1. *It is not clear the connection between PCM and CML. Which is the rationale for choosing CML as a model of PCM? This must be addressed more in detail, because to my knowledge PCM has not been described for this disease.*

We appreciate the reviewer's comment and agree that the link between CML and PCM is not intuitively obvious. To explain this in more detail, we would note that our lab is primarily focused on transplantation immunology with a specific emphasis on the biology of graft versus host disease (GVHD) and graft versus leukemia (GVL) responses that occur during allogeneic hematopoietic stem cell transplantation. The CML mouse model described in this manuscript is specifically used in these studies as a readout for assessing GVL responses. During these experiments, it is also customary to examine the primary GVHD target organs which are the liver, lung, and colon to discern whether strategies to enhance a GVL effects deleteriously promote GVHD. As we conducted these studies, we unexpectedly observed that PCM arose in mice that served as the leukemia controls for these experiments. We did not specifically state this in the manuscript since it requires a rather lengthy explication that was outside the main scope of the paper. Rather, we presented this in a more abbreviated fashion, noting that this was an unexpected finding during our histopathological examination of the colon during these studies.

2. *Authors should state clearly that off-tet is CML and to use the same nomenclature in all text.*

We have modified the text so that the nomenclature is now standardized throughout the manuscript, specifically within the figures.

3. *Discussion line 306: Authors stated, "to validate a causative relationship between PCM and CML, we employed a tet mouse model" The causative relationship is still not clear, as example how BCR: ABL1 is inducing PCM. Authors are describing an association. Authors should try to engraft mouse with primary cells from CML patients and verify PCM.*

We regret that we were imprecise with the words we used to describe the connection between CML and PCM. Our intention was to emphasize the linkage between these two processes, not that bcr/abl oncogene directly caused PCM. We have therefore modified the sentence to state "To validate a direct linkage between CML and PCM...". With respect to the engraftment of immunocompetent mice with primary cells from CML patients, we believe that this would be problematic for several reasons. The first is that white blood cells from CML patients are largely mature granulocyte populations that have no potential for self-renewal; therefore, they cannot recapitulate CML. Secondly, transplantation of these cells would likely generate a host anti-donor response which would result in the immune-mediated eradication of the cells.

4. *It is not clear the translation value of this value for CML patients.*

We agree with the reviewer that there is no obvious translational value for CML patients. This is due in large measure to the fact that it is not ethically possible to confirm this association in human CML patients. The diagnosis of CML is made by performing a bone marrow aspirate and biopsy, whereas determination of associated PCM would require a coincidental colonoscopy and biopsy of colonic tissue. A concurrence of these clinical events virtually never occurs, and it would be unethical to perform a colonoscopy for the sole purpose of documenting the presence or absence of PCM if there was no specific gastrointestinal tract-related question that required assessment. We would, however, emphasize that this was not the primary goal of the manuscript. Rather, our intent was to provide a mechanistic explanation for how PCM develops since the pathophysiology of this entity has long been poorly understood. Unexpectedly, the vehicle for dissecting this pathway proved to be a murine model of CML where we observed the development of PCM coincident with disease onset. Thus, we believe that a lack of apparent translational value does not negate the novel mechanistic insights provided by this murine model as it provides new insights into the pathophysiology of PCM and also elucidates a heretofore unappreciated IL-33/ILC2/IL-9 immune circuit that induces metaplasia and intestinal remodeling.

5. *This mouse model is a very specific model driven by BCR: ABL1, and this pathway should be validated in another model, for example in intestinal organoids in a model without BCR: ABL1.*

We extensively reviewed the literature to determine whether there were any other disease specific murine models in which PCM was reported to occur. This review process did not reveal any other disease-oriented models which describe this phenomenon, and therefore would allow us to validate this mechanistic pathway in another setting. We therefore know of no way to address this query. We believe, however, that the absence of any other existing models is an innovative aspect of the paper since it defines for the first time a mechanistic pathway by which PCM develops. In addition, this manuscript now provides important foundational data for future studies wherein another model of PCM is discovered and can then be examined to validate the relevance of the pathways that we have delineated in the current report.

6. *Because of the model used, it is still not totally clear the translation value in the context of CML, especially following author's conclusions stating "IL-33-regulated ILC2/IL-9 immune circuit... may have pathophysiological relevance for the emergence of PCM in diseases as colorectal cancer and inflammatory bowel disease" at line 416; however, this has not been proven on these settings. Additional evidence is needed.*

We agree with that reviewer that we have not proven that the mechanistic pathway delineated in the current report will be valid in other diseases, particularly those in the gastrointestinal tract. We have therefore modified the last sentence of the manuscript to indicate that future studies are required to determine the pathophysiological relevance of this pathway in these other settings.

REVIEWER COMMENTS

Reviewer #1 (Remarks to the Author):

1) New IF data (Lyz, Fig.1E) is not well-correlated to Fig.1F-G (H&E) and Fig.1H-I (QPCR), which will raise concerns about the extent of PCM in CML mice. There are good IF antibodies (Lyz, MMP7, MPTX1/2) to specifically track Paneth cells in the intestine. There is a good Lyz-FITC Ab (Lysozyme EC 3.2.1.17, Dako) for flow cytometry to track Paneth cells by %. We regularly performed Lyz IF staining for the ileum and Fig.1E ileum CML is not really a PCM phenotype to me. With this concern, additional IF data for Paneth cells is also needed to further justify Fig.2C-F, regarding "IL-9 promotes PCM (or anti-IL-9 blocks PCM)", as this is a key conclusion in this study (title). Furthermore, if the model is correct, anti-IL-33/ST2 blockade will also have an inhibitory effect on PCM (related to Fig.8L-N), where clear IF data for Paneth cells should also be provided.

2) Based on the model, IL-9 antibody blocking experiments (IL-33/ST2 blocking is essentially important as well, but less important with IL-13/IL-5 or IL-25 blockade) are key to this study, since this is the key mechanistic study. Therefore, results of IL-9 blockade, in the context of CML, in the axis of CML-IL-33-ILC2-IL-9 -> PCM (or -> mast cell loop) should be convincing to support the main conclusion and novelty. Again, while all other findings (IL-33-ILC2-IL-9, Mast-IL-33 cleavage, IL-9-producing ILC2, IL-9 -> Paneth cell hyperplasia in *J Immunol* 182:4737) have been previously reported, the authors would want to distinguish their study from previous work by showing a) the importance or the role of CML in the circuit (not mentioned in the title) or, as suggested by other reviewers, b) to find a second PCM model to demonstrate the unique (or universal I would say) role of IL-9 in PCM in the absence of CML.

First, I was a little bit surprised to see that this CML model did not cause any detectable colon pathology (Fig.7A), but this could be due to the fact that CML mice in authors' exp setting (I don't know if this could be adjusted by transferring more BM cells into donor to increase the severity of inflammation or PCM) usually lose mild 10% of body weight (Fig.S2A), which unlikely will cause colon pathology based on our experience in DSS colitis model. However, more unexpectedly, instead of ameliorating inflammation, global IL-9 blockade in CML mice cause significantly more body weight loss (~20%) and an increase in granulocyte expansion in peripheral blood and spleen (Fig.S2). I am wondering at the 20% weight loss after IL-9 blockade, those anti-IL-9-treated CML mice might have shown some gut pathology. While the authors claimed that this abnormality is beyond the scope of study, this result somewhat reflects a conflict to the model in Fig.9, where IL-9 promotes mast cell/neutrophils (so the IL-9 blockade should inhibit this at least locally). With a side effect of IL-9 blockade on granulopoiesis (or likely causing mast cell expansion), those exp in this study with this global anti-IL-9 treatment may bring additional concerns in terms of data interpretation (Fig.2C-F, 4K-S, 8F-K). In the worse case, IL-9 blockade may even increase the severity of CML (myeloid expansion and proinflammatory condition). Regarding this, the authors did not fully address my previous concerns (Fig.S2B-E were done in the periphery and spleen but not colon lamina propria like Fig.7B, where the regulatory events take place in the intestine).

To be specific, do the authors think that the (IL-9-mast cell-IL-33) loop plays a role in (CML-epithelial damage-IL-33) pathway? That was partially shown in Fig.8F-G, at the downstream bio-active level of IL-33. But whether or not IL-9 blockade can inhibit IL-9-mast cell loop, in the context of CML, needs additional data (Flow analysis of mast cell/neutrophil/macrophage after IL-9 blockade, like Fig.7B). This will also provide clues to distinguish (CML -> epithelial damage -> IL-33-ILC2-IL-9 -> mast cell loop -> IL-33) from (CML -> mast cell expansion -> IL-33-ILC2-IL-9 -> mast cell loop). If CML could predominantly trigger mast cell expansion in the gut (2nd case), then IL-9 blockade may not effectively block mast cell -> IL-33 loop. In contrast, if IL-33 triggers mast cell loop (1st case), IL-9 blockade will not ameliorate CML -> epithelial damage. At this point, it is still unclear to me about the CML-ILC2 circuit, plus, the side effect of IL-9 blockade on CML would add more complexity to the model. For this reason, it may be justified for the reviewers to ask for a second PCM model in the absence of CML, where the authors can use unbiased IL-9 blockade to test the underlying mechanism (i.e. IL-9 blockade will not have a side effect on granulopoiesis in the context of CML).

3) In Fig.7E. high E-Cad signals at the crypt base in control ileum is abnormal, which is not seen in

CML ileum or colon sections. IF data quality needs to be improved. Also, Fig.7E may not support the IEC apoptosis (E-Cad stain is on the membrane while caspase-3 is mostly in the nucleus, if confocal resolution is not high, one would not see caspase-3 signals within E-Cad signals since these two signals by compartments should not be col-localized). Therefore, the authors should confirm that samples used in Fig.7F were purified ileum/colon IEC but not ileum/colon fragments in order to justify epithelial damage.

Reviewer #2 (Remarks to the Author):

The authors have addressed all my comments and concerns successfully.

Reviewer #3 (Remarks to the Author):

The authors have addressed all the reviewers's suggestions, therefore I recommend to accept it in the following format.

Minor comment: at page 4 Results, there is not need to write CML in full because the name already appears few lines up.

The manuscript is not clearer and highlights several important new findings: 1- presence of inflammation in the CML gastrointestinal tract; 2- CML intestinal remodelling; 3- determination of the mechanistic pathway for the development of PCM. This highlights possible new therapeutic targets which could be helpful for patients.

The comments of Reviewer 1 were addressed as follows:

1. *New IF data (Lyz, Fig.1E) is not well-correlated to Fig.1F-G (H&E) and Fig.1H-I (QPCR), which will raise concerns about the extent of PCM in CML mice. There are good IF antibodies (Lyz, MMP7, MPTX1/2) to specifically track Paneth cells in the intestine. We regularly performed Lyz IF staining for the ileum and Fig.1E ileum CML is not really a PCM phenotype to me. With this concern, additional IF data for Paneth cells is needed. With this concern, additional IF data for Paneth cells is also needed to further justify Fig.2C-F, regarding “IL-9 promotes PCM (or anti-IL-9 blocks PCM).*

As requested, we have performed additional immunofluorescence (IF) imaging and employed another lysozyme antibody (total of two; Clones EPR2994(2) and BGN/0696/5B1) which demonstrates more robust staining of Paneth cells in the ileum of CML mice. In addition, we have conducted staining with MPTX 1/2 which shows a significantly increased number of Paneth cells in the ileum of these animals. We would add that, in conjunction with H & E staining, tartrazine staining, electron microscopy, and qPCR results for defensins, we believe we have compelling data with a total of six different approaches that demonstrate the presence of Paneth cell metaplasia in the colon and Paneth cell hyperplasia in the ileum. The lysozyme images are now depicted as new **Figure 1E**. MPTX 1/2 images are shown as new **Supplemental Figure 2**. We also provide IF images that demonstrate that anti-IL-9 antibody administration results in the reduction of PCM in the colon of CML mice, but not of Paneth cell hyperplasia in the ileum, which is consistent with data presented in Figure 2C. These data are depicted as new **Figure 2E**.

2. *Based on the model, IL-9 antibody blocking experiments (IL-33/ST2 blocking is essentially important as well, but less important with IL-13/IL-5 or IL-25 blockade) are key to this study, since this is the key mechanistic study. Therefore, results of IL-9 blockade, in the context of CML, in the axis of CML-IL-33-ILC2-IL-9 -> PCM (or -> mast cell loop) should be convincing to support the main conclusion and novelty. Again, while all other findings (IL-33-ILC2-IL-9, Mast-IL-33 cleavage, IL-9-producing ILC2, IL-9 -> Paneth cell hyperplasia in J Immunol 182:4737) have been previously reported, the authors would want to distinguish their study from previous work by showing a) the importance or the role of CML in the circuit (not mentioned in the title) or, as suggested by other reviewers, b) to find a second PCM model to demonstrate the unique (or universal I would say) role of IL-9 in PCM in the absence of CML.*

With respect to the effects of IL-9 antibody blockade on the IL-33→ILC2→IL-9→PCM/intestinal remodeling immune circuit, we believe that our results strongly support the pivotal and critical role of IL-9 in mediating this process. Specifically, our data demonstrate that blockade of IL-9 signaling abrogates the development of PCM and Paneth cell hyperplasia (**Figures 2C-2G**), significantly reduces intestinal remodeling as characterized by tuft cell hyperplasia (**Figures 4L-4O**), goblet cell hyperplasia (**Figures 4P-4S**), and mast cell hyperplasia (**Figures 4B-4F**), and significantly decreases protein expression of neutrophil elastase (**Figure 8I**), cathepsin G (**Figure 8J**), mMCP1 (**Figure 8K**), and the more potent short form of IL-33 (**Figure 8H**). Thus, our results demonstrate that all of these elements of this immune circuit are regulated by IL-9. As we point out below in more detail, blockade of IL-9 has no effect on decreasing expression of the long form of IL-33 and by extension epithelial cell damage, but only the short form which initiates all of the events detailed above. Thus, we believe that these data convincingly demonstrate that IL-9 plays a critical role in the regulation of the downstream events of PCM and intestinal remodeling in this immune circuit.

The reviewer also drew a parallel between data presented in our manuscript and the prior publication by Steenwinckel and colleagues (*J Immunol*, 182, 2009), implying that data presented

in our report merely recapitulated that previously published by Steenwinckel, and therefore required a second PCM model to establish novelty. However, a reexamination of this earlier publication reveals that these authors only demonstrated that IL-9 induces PCM in IL-9 transgenic animals and provided no investigation of upstream pathways or downstream effects on small intestinal remodeling. Moreover, they provided no evidence that blockade of this pathway abrogated PCM to any extent. In contrast, our paper provides a much more detailed analysis which delineates a mechanistic pathway by which intestinal epithelial damage in the setting of a CML-induced proinflammatory environment induces IL-33 production which results in the activation of type 2 ILCs followed by the production of IL-9. IL-9 then induces downstream intestinal remodeling which is characterized by goblet cell, tuft cell, and mast cell hyperplasia. Furthermore, our manuscript validates a pivotal role for IL-9, by showing that blockade of this cytokine reverses PCM and intestinal remodeling. Finally, we also formally exclude a role for other TH2 cytokines (i.e., IL-5) and cytokines that have been implicated in intestinal remodeling in other murine models (i.e., IL-25). Thus, we believe that our manuscript details a novel mechanistic pathway that is much more comprehensive and extends far beyond what was reported in the prior Steenwinckel publication.

While the authors claimed that this abnormality is beyond the scope of study, this result somewhat reflects a conflict to the model in Fig.9, where IL-9 promotes mast cell/neutrophils (so the IL-9 blockade should inhibit this at least locally). In the worst case, IL-9 blockade may even increase the severity of CML (myeloid expansion and proinflammatory condition). Regarding this, the authors did not fully address my previous concerns (Fig.S2B-E were done in the periphery and spleen but not colon lamina propria like Fig.7B, where the regulatory events take place in the intestine).

The reviewer noted that administration of anti-IL-9 antibody resulted in granulocytic hyperplasia in the spleen and queried whether similar findings, such as mast cell hyperplasia and neutrophil hyperplasia, occurred locally in the colon. In response to this query, we would point out that we did conduct studies that demonstrated that anti-IL-9 antibody signaling blockade resulted in a significant reduction in protein expression of neutrophil elastase (**Figure 8I**) and cathepsin G (**Figure 8J**) in the colon, and mMCP1 in both the ileum and colon (**Figure 8K**). Since neutrophil elastase and cathepsin G are expressed in neutrophils, and mMCP1 is expressed only in mast cells, our data demonstrate that anti-IL-9 antibody administration significantly decreases neutrophil and mast cell accumulation locally in the colon of CML mice. Thus, blockade of IL-9 does in fact reduce accumulation of these cells in the GI tract in contrast to what is observed in the spleen.

To be specific, do the authors think that the (IL-9-mast cell-IL-33) loop plays a role in (CML-epithelial damage-IL-33) pathway? That was partially shown in Fig.8F-G, at the downstream bio-active level of IL-33. But whether or not IL-9 blockade can inhibit IL-9-mast cell loop, in the context of CML, needs additional data (Flow analysis of mast cell/neutrophil/macrophage after IL-9 blockade, like Fig.7B). This will also provide clues to distinguish (CML -> epithelial damage -> IL-33-ILC2-IL-9 -> mast cell loop -> IL-33) from (CML -> mast cell expansion -> IL-33-ILC2-IL-9 -> mast cell loop). If CML could predominantly trigger mast cell expansion in the gut (2nd case), then IL-9 blockade may not effectively block mast cell -> IL-33 loop. In contrast, if IL-33 triggers mast cell loop (1st case), IL-9 blockade will not ameliorate CML -> epithelial damage. At this point, it is still unclear to me about the CML-ILC2 circuit, plus, the side effect of IL-9 blockade on CML would add more complexity to the model.

The reviewer has requested clarification of the proposed model which we present in Figure 9. With respect to the individual points/questions raised above, we would make the following points. First of all, regarding the role of IL-9 in inhibiting mast cell hyperplasia, our data indicate that blockade

of IL-9 signaling significantly reduces both gene (**Figure 4C**) and protein (**Figure 8K**) expression of the mast cell specific gene, mMCP1. Thus, inhibition of IL-9 significantly attenuates mast cell hyperplasia. We would also note that mast cells are present only in the epithelium and not in the lamina propria (**Figure 4E**) so therefore cannot be assessed as in Figure 7B which examined innate immune cells in the lamina propria. With respect to whether IL-33 can “trigger” mast cells, our data as shown in **Figure 8N** demonstrate that administration of anti-ST2 antibody has no effect on the expression of mMCP1, indicating that there is no direct effect of IL-33 on mast cells. Also, we would point out that blockade of IL-9 does not reduce the long form (35 kDa) of IL-33 but only the short form (15 kDa) which is the direct result of protease cleavage. This is likely due to the fact that IL-9 induces mast cell hyperplasia which leads to release of mMCP1 and subsequent IL-33 cleavage. Consequently, IL-9 blockade does not prevent IL-33 release, but only the cleavage into a more potent biologically active form. Thus, these data indicate that IL-9 does not play a direct role in CML-induced epithelial cell damage nor in the release of the long form of IL-33 which is released by damaged epithelial cells.

Therefore, in summary, we believe that our data support a model that is predicated on the release of IL-33 from damaged epithelial cells. IL-33 is then cleaved into its more potent short form as a result of proteases released predominantly by mast cells, although we cannot exclude a contribution from neutrophil proteases as well. This short form of IL-33 then binds to ST2 on type 2 ILCs resulting in the production of IL-9, which is the dominant cytokine that drives PCM, mast cell, tuft cell and goblet cell hyperplasia. This is corroborated by the fact that administration of anti-IL-9 antibody results in a significant reduction in PCM, hyperplasia of specialized intestinal populations, and the short, more biologically active form of IL-33.

- 3. In Fig.7E, high E-Cad signals at the crypt base in control ileum is abnormal, which is not seen in CML ileum or colon sections. IF data quality needs to be improved. Also, Fig.7E may not support the IEC apoptosis (E-Cad stain is on the membrane while caspase-3 is mostly in the nucleus, if confocal resolution is not high, one would not see caspase-3 signals within E-Cad signals since these two signals by compartments should not be col-localized). Therefore, the authors should confirm that samples used in Fig.7F were purified ileum/colon IEC but not ileum/colon fragments in order to justify epithelial damage.*

We regret that the immunofluorescence (IF) images showing both E-cadherin and cleaved caspase 3 may not have been sufficiently clear. To that end, we have now recut tissue blocks and re-stained them with a different E-cadherin antibody (Clone 4A2) so that Figure 7E no longer shows a high signal at the base of the ileum. This is now depicted in new **Figure 7E**. Furthermore, we provide high-power images which clearly show that E-cadherin and cleaved caspase 3 do not co localize in epithelial cells (**new Supplemental Figure 6**). The formal proof for this conclusion is that E-cadherin is labeled as green in IF images, whereas cleaved caspase 3 is labeled as red. If these two proteins co-localized within the same compartment of the cells, then co localization would appear as yellow in the colon and ileum samples. However, this does not occur as high-power images demonstrate separation of both colors in the tissue sections with cleaved caspase in the nucleus and E-cadherin outside the nucleus in the cytoplasm. In contrast, DAPI which labels blue and cleaved caspase 3 which labels red do co-localize in the nucleus and thereby show as pink-appearing cells, verifying nuclear localization. Thus, E-cadherin and cleaved caspase 3 do not co-localize in the epithelium of the CML mice and are present in different cellular compartments.

We are gratified that we have addressed all the concerns of Reviewer 2 and that the paper is deemed to be acceptable for publication.

The minor comment of Reviewer 3 was addressed as follows:

- 1. On page 4 of the Results section, there is no need to write CML in full because the name already appears a few lines up.*

We have employed the abbreviation for CML as requested.

Otherwise, Reviewer 3 deemed the manuscript acceptable for publication.

We thank Reviewer 3 for his acknowledgment that our manuscript highlights several important new findings; specifically, (1) the presence of inflammation in the CML gastrointestinal tract; (2) the development of intestinal remodeling in this disease; and (3) the determination of the mechanistic pathway by which PCM develops in the GI tract. Furthermore, the reviewer notes that our work provides possible new therapeutic targets that could have translational relevance in patients.

REVIEWERS' COMMENTS

Reviewer #1 (Remarks to the Author):

Comments for Author:

I appreciate and are satisfied with the authors' efforts in providing new data and appropriate response to my previous concerns. I agree that the manuscript is in a good shape for publication.